# Functional and structural insights into the multi-step activation and catalytic mechanism of bacterial ExoY nucleotidyl cyclase toxins bound to actin-profilin

**Magda Teixeira Nunes[1], Pascal Retailleau[2], Dorothée Raoux-Barbot[3], Martine Comisso[1], Anani Amegan Missinou[1], Christophe Velours[1¤], Stéphane Plancqueel[1], Daniel Ladant[3], Undine Mechold[3], Louis Renault[1] ***

**1** Université Paris-Saclay, CEA, CNRS, Institute for Integrative Biology of the Cell (I2BC), Gif-sur-Yvette, France, **2** Université Paris Saclay, CNRS, Institut de Chimie des Substances Naturelles, Gif-sur-Yvette, France, **3** Institut Pasteur, Université Paris Cité, CNRS UMR 3528, Unité de Biochimie des Interactions macromoléculaires, Département de Biologie Structurale et Chimie, Paris, France

¤ Current address: Fundamental Microbiology and Pathogenicity Laboratory, UMR 5234 CNRS-University of Bordeaux, SFR TransBioMed, Bordeaux, France
* louis.renault@i2bc.paris-saclay.fr

**Data Availability Statement:** Structural data have been deposited in the Protein Data Bank and will be

## Abstract

ExoY virulence factors are members of a family of bacterial nucleotidyl cyclases (NCs) that are activated by specific eukaryotic cofactors and overproduce cyclic purine and pyrimidine nucleotides in host cells. ExoYs act as actin-activated NC toxins. Here, we explore the *Vibrio nigripulchritudo* Multifunctional-Autoprocessing Repeats-in-ToXin (MARTX) ExoY effector domain (Vn-ExoY) as a model for ExoY-type members that interact with monomeric (G-actin) instead of filamentous (F-actin) actin. Vn-ExoY exhibits moderate binding affinity to free or profilin-bound G-actin but can capture the G-actin:profilin complex, preventing its spontaneous or VASP- or formin-mediated assembly at F-actin barbed ends *in vitro*. This mechanism may prolong the activated cofactor-bound state of Vn-ExoY at sites of active actin cytoskeleton remodelling. We present a series of high-resolution crystal structures of nucleotide-free, 3'-deoxy-ATP- or 3'-deoxy-CTP-bound Vn-ExoY, activated by free or profilin-bound G-actin-ATP/-ADP, revealing that the cofactor only partially stabilises the nucleotide-binding pocket (NBP) of NC toxins. Substrate binding induces a large, previously-unidentified, closure of their NBP, confining catalytically important residues and metal cofactors around the substrate, and facilitating the recruitment of two metal ions to tightly coordinate the triphosphate moiety of purine or pyrimidine nucleotide substrates. We validate critical residues for both the purinyl and pyrimidinyl cyclase activity of NC toxins in Vn-ExoY and its distantly-related ExoY from *Pseudomonas aeruginosa*, which specifically interacts with F-actin. The data conclusively demonstrate that NC toxins employ a similar two-metal-ion mechanism for catalysing the cyclisation of nucleotides of different sizes. These structural insights into the dynamics of the actin-binding interface of actin-activated ExoYs and the multi-step activation of all NC toxins offer new perspectives for the specific inhibition of class II bacterial NC enzymes.

available upon publication under PDB codes: 8BJH, 8BJI, 8BJJ, 8BR1, 8BO1, 8BR0. See S2 Table for further details. https://doi.org/10.2210/pdb8BR0/pdb https://doi.org/10.2210/pdb8BJI/pdb https://doi.org/10.2210/pdb8BJH/pdb https://doi.org/10.2210/pdb8BJJ/pdb https://doi.org/10.2210/pdb8BR1/pdb https://doi.org/10.2210/pdb8BO1/pdb.

**Funding:** This work was supported by the ANR under ANR-18-CE44-0004 (to L.R., M.T.N., P.R., U.M.), by CNRS (to L.R., P.R., U.M., D.L.) and Institut Pasteur under #PTR 43-16 (to U.M.). The funders had no role in study design, data collection and analysis, decision to publish, or preparation of the manuscript.

## Author summary

ExoY toxins are bacterial nucleotidyl cyclases (NCs) injected into eukaryotic cells, binding to specific host cofactors to trigger their toxic, potent NC enzymatic activity. They alter host cell signalling by overproducing purine and pyrimidine cyclic nucleotides, which serve as canonical and non-canonical intracellular messengers, respectively. The molecular and mechanistic details underlying the activation and catalytic specificities of NC toxins are only partially understood. In this study, we examine ExoY-like NCs exclusively activated by monomeric forms of actin, in contrast to other family members activated through interactions with actin filaments. Our *in vitro* investigations reveal that these ExoYs capture the actin:profilin complex to trigger activation, disrupting its association with the most dynamic ends of actin filaments. We present two structural snapshots along the Vn-ExoY activation pathway by G-actin or G-actin-profilin without or with purine or pyrimidine nucleotide analogues. Our structural data unveil unprecedented mechanistic details of how the active site of all NC toxins undergoes sequential remodelling upon cofactor and substrate binding, how they can accommodate nucleotides of different sizes as substrates, and elucidate crucial features of their catalytic reaction. These structural insights into the multi-step activation of NC toxins open new avenues to inhibit specifically this class of toxic bacterial NCs.

## Introduction

ExoY-like nucleotidyl cyclase (NC) toxins are virulence factors produced by several Gram-negative β- and γ-Proteobacteria [1]. They belong to a family of bacterial NCs that share a structurally-related catalytic NC domain. Whereas these NCs are inactive inside bacteria, their enzymatic activity is significantly stimulated upon injection into eukaryotic cells by binding to specific eukaryotic proteins [2,3]. This turns them into potent cyclases that can catalyse the production of substantial amounts of cyclic purine (cAMP, cGMP) and pyrimidine (cCMP, cUMP) nucleotides [4,5], thus interfering with host cell signalling (reviewed in [1,2,6–8]). While the *Bacillus anthracis* edema factor (EF) and *Bordetella pertussis* CyaA toxins are both activated by calmodulin (CaM), the ExoY-like NC toxins utilise actin [3]. Extensive research has been conducted on CaM-activated toxins over the past decades. In contrast, the functional characteristics and virulence mechanisms of the ExoY-like NC toxins still need to be clarified.

Exoenzyme Y (Pa-ExoY) is the second most prevalent type III secretion system (T3SS) exotoxin encoded in the genomes of clinical or environmental isolates of *P. aeruginosa* [9]. Pa-ExoY's NC activity in host cells alters the host immune response [10,11], leads to the release of cytotoxic amyloids from the pulmonary endothelium and impairs cellular repair after infection [7,12,13]. In *Vibrio vulnificus* biotype 3, a pathogen causing severe foodborne and wound infections in humans, the ExoY module of its MARTX toxin (Vv-ExoY) is essential for virulence [14]. ExoY NC homologues from different pathogens can potentially exert different cytotoxic effects in infected cells due to differences in their protein sequence, the form of actin they require to become active, and their substrate specificities [5,15]. The MARTX Vn-ExoY module of *Vibrio nigripulchritudo*, an emerging pathogen in marine shrimp farming, is representative of the ExoY modules found in *Vibrio* strains. It shares only 38% sequence similarity with Pa-ExoY. While Pa-ExoY requires actin filaments (F-actin) for maximal activation [3], Vn-ExoY is selectively activated by G-actin *in vitro*. Vn-ExoY, like EF and CyaA [16], strongly prefers ATP and can use CTP as substrate, but much less efficiently [5]. Pa-ExoY, on the other hand, synthesises a wide range of cyclic nucleotide monophosphates (cNMPs) [16], with the

following *in vitro* substrate preference: GTP>ATP≥UTP>CTP [5]. Recently, cUMP and cCMP have emerged as novel second messengers that activate antiviral immunity in bacteria [17]. Their role in eukaryotic cells, however, remains largely unclear [18].

Bacterial NC toxins are classified as class II adenylate cyclases (ACs), distinguishing them from the widely distributed class III adenylyl or guanylyl cyclases (GCs) [19–26]. In their cofactor-bound state, they are several orders of magnitude more active than activated mammalian class III ACs [2], which use a dimeric catalytic architecture. The molecular basis for the turnover differences between these two AC classes remains to be elucidated. The catalytic domain of NC toxins can be divided into subdomains $C_A$ and $C_B$, with the catalytic site located in a groove at their interface [24]. EF and CyaA recognise CaM through flexible regions of $C_A$ termed switch A (residues 502–551 in EF, 199–274 in CyaA) and switch C (630–659 in EF, 348–364 in CyaA) [24,25,27]. In CaM-bound EF, switch A and C stabilise another key region of $C_A$ called switch B (578–591 in EF, 299–311 in CyaA), which contains residues that either directly bind ATP or stabilise catalytic residues in the $C_A$-$C_B$ groove [24]. When EF is inactive, switch A and C are distant from switch B, which is disordered. Activation of EF or CyaA by CaM does not appear to require a large subdomain movement between $C_A$ and $C_B$, but rather a CaM-induced movement of switch A and C towards the catalytic site [2]. This in turn stabilises switch B and ATP binding to favour a catalytically competent form of the enzyme. This allosteric activation-by-stabilisation mechanism has recently gained support from studies on actin-activated NC toxins. In the case of free Pa-ExoY crystallised after limited in-situ proteolysis, the three switches are not discernible in the electron density, either due to partial proteolysis and/or flexibility [26]. However, recent cryo-EM structures of actin-activated Pa-ExoY and Vv-ExoY have elucidated the involvement of their central switch A and C-terminal switch C in recognising F- and G-actin subunits, respectively [28]. Molecular dynamics simulations propose that the flexible switches A and C of free Pa-ExoY become stabilised upon binding to F-actin, thereby conferring stability to the entire toxin and its nucleotide-binding pocket (NBP). Despite these advances, the interactions with ligands in the active sites of CaM- or actin-activated NC toxins remain heterogeneous. The optimal number of ions required, the nature of the catalytic base, and the coordination and role of residues involved in various reaction steps are subject of ongoing debate [1,24,25,27–33]. Furthermore, the molecular basis of the NC toxins' pyrimidinyl cyclase activity remains unknown. The present study aims to shed light on this intriguing aspect.

Through *in vitro* functional and structural studies, we demonstrate that Vn-ExoY seizes the actin:profilin complex for activation by disrupting its association with the most dynamic ends (barbed-ends) of actin filaments. Our work includes a series of high-resolution structures of Vn-ExoY, activated by free or profilin-bound actin-ATP/ADP, both in the presence and absence of non-cyclisable ATP or CTP analogues bound to $Mg^{2+}$ or $Mn^{2+}$ ions. These structural findings unveil essential features of the interface and functional dynamics of ExoY NC toxins with G/F-actin, elucidating how NC toxins accommodate large purine or small pyrimidine nucleotides (with double- or single-carbon nitrogen ring bases, respectively) as substrates. Furthermore, we validate the significance of specific residues involved in the purinyl and pyrimidinyl cyclase activity of NC toxins in both Vn-ExoY and the distantly-related Pa-ExoY.

## Results

### Vn-ExoY binds to actin monomers with only modest affinity and inhibits actin self-assembly

As a model for ExoY toxins that are selectively activated by G-actin, we used a functional Vn-ExoY$^{3412-3873}$ effector domain/module corresponding to residues 3412 to 3873 of *V. n.*

MARTX [5]. For ease of reading and illustration, Vn-ExoY residue numbers are shown subsequently without thousands. For example, *V.n.* MARTX residue 3412 is now 412. We first examined the effect of Vn-ExoY on actin-ATP/-ADP self-assembly. To avoid the indirect effects of ATP depletion through Vn-ExoY AC activity, we used an inactive double mutant K528M/K535I, hereafter referred to as Vn-ExoY$^{DM}$ (corresponding to Pa-ExoY$^{K81M/K88I}$) [5]. Actin polymerisation kinetics were monitored using pyrene-labelled actin, which shows an increase in fluorescence intensity when incorporated into filaments. While inactive Pa-ExoY$^{K81M}$ accelerated the rate of actin polymerisation (Fig 1A) [28], Vn-ExoY$^{DM}$ inhibited the polymerisation rate of both G-actin-ATP-Mg and G-actin-ADP-Mg (Fig 1B and 1C). Thus, Vn-ExoY interacts with both G-actin-ATP or -ADP, but in a very different way to Pa-ExoY. To understand the inhibitory effect of Vn-ExoY, we examined the association of free or Vn-ExoY$^{DM}$-bound G-actin with the barbed (+) or pointed (-) ends of filaments. Spectrin-actin or gelsolin-actin seeds were used to induce elongation at the barbed- and pointed-ends, respectively. Interaction of Vn-ExoY$^{DM}$ with G-actin-ATP-Mg inhibited actin assembly at both ends

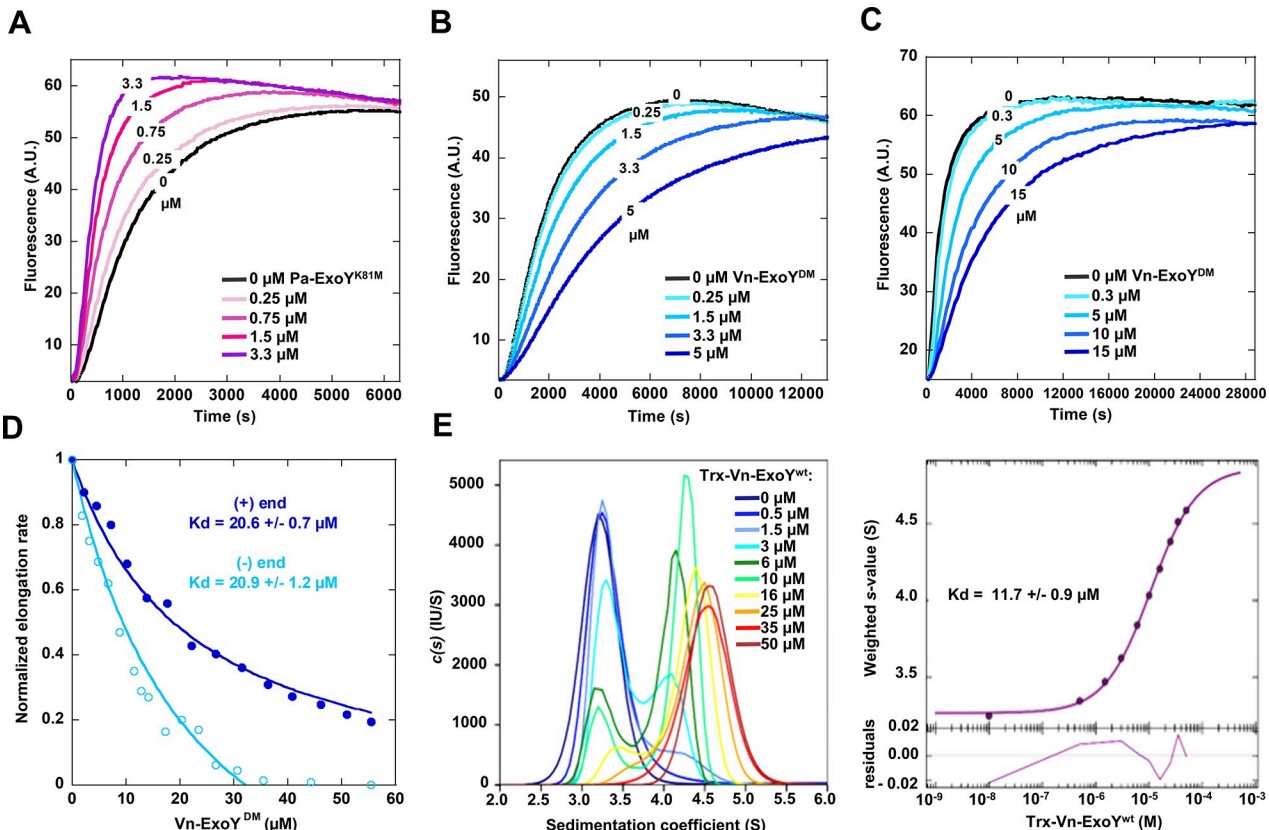

**Fig 1. Vn-ExoY sequesters actin-ATP/ADP monomers in actin self-assembly by interacting with modest affinity.** (A-C) Polymerisation time course of 4 μM ATP-Mg-G-actin (5% pyrenyl-labelled, (A-B)) or 10 μM ADP-Mg-G-actin (10% labelled, (C)) in the absence (black) and presence of Pa-ExoY$^{K81M}$ (A) or Vn-ExoY$^{DM}$ (B-C) at the indicated concentrations (μM). (D) Initial rates of barbed-end (dark blue) or pointed-end (light blue) growth from spectrin-actin or gelsolin-actin seeds, respectively, were measured with 1.5 μM G-actin (10% pyrenyl-labelled) and increasing concentrations of Vn-ExoY$^{DM}$. Initial rates were normalised to the initial elongation rate with free G-actin from a linear fit to the first 120 s of each time course. Vn-ExoY$^{DM}$ inhibits both barbed and pointed-end growth by sequestering G-actin with the indicated Kd (μM). (E) AUC-FDS measurement of the binding affinity of Vn-ExoY for G-actin. The sedimentation coefficient distribution c(s) (left panel) of Alexa-488-labelled-G-actin-ATP (60 nM) bound to the actin-polymerisation inhibitor Latrunculin A (2.5 μM) was determined in the absence and presence of increasing concentrations of unlabelled Trx-Vn-ExoY (from 0.5 to 50 μM). The resulting isotherms of weight-average s values from c(s) integration are shown in the right panel for the interaction between Trx-Vn-ExoY and G-actin-ATP-Latrunculin-A according to a 1:1 model. In the isotherm plots, the solid circles are the sw data from the dilution series. The solid line is the best-fit isotherm giving an estimated Kd of 11.7 μM for the interaction between Trx-Vn-ExoY and G-actin-ATP-Latrunculin-A.

(Fig 1D). The dose-dependent effects of Vn-ExoY$^{DM}$ on barbed- and pointed-end elongation rates were consistent with the formation of a 1:1 molar complex between G-actin and Vn-ExoY$^{DM}$ sequestering G-actin with an equilibrium dissociation constant (Kd) of 21 μM (Fig 1D). To confirm the binding affinity by direct measurement, we performed fluorescence-detected sedimentation velocity experiments using analytical ultracentrifugation (AUC-FDS) with G-actin-ATP labelled with the fluorescent dye Alexa-488 and unlabelled Vn-ExoY fused to Thioredoxin (Trx-Vn-ExoY). This protein fusion, more soluble than Vn-ExoY, was preferred for high concentration dose-response analyses. With free labelled G-actin (0.06 μM) bound to Latrunculin-A, an actin-polymerisation inhibitor, the actin sedimentation coefficient distribution gave a peak centred at an $s_{20,w}$ value of 3.3 S (Fig 1E, left panel) consistent with a globular actin monomer of 42 kDa. Upon incremental addition of Trx-Vn-ExoY, the sedimentation coefficient distribution of labelled actin shifted to higher s-values (from 3.2 up to 4.7 S), demonstrating the interaction of Vn-ExoY with actin. Peak position and shape were concentration-dependent, and the relative proportion of the larger species in the peaks increased with increasing Trx-Vn-ExoY concentrations. The titration was consistent with the formation of a 1:1 Vn-ExoY:G-actin complex with a Kd of 11.7 ± 0.9 μM (Fig 1E, right panel). Vn-ExoY therefore forms a sequestering but modest affinity complex with G-actin. Its affinity is modest compared to that of Pa-ExoY for F-actin [3] or EF and CyaA for calmodulin [34,35] (S1 Table). It is also modest compared to that of G-actin binding proteins (G-ABPs), which regulate the G-actin pool in eukaryotic cells [36] and reduce the concentration of free G-actin to prevent uncontrolled actin nucleation.

## Vn-ExoY is efficiently activated by G-actin bound to profilin

We next examined the *in vitro* AC activity of Vn-ExoY stimulated by G-actin in the presence of three regulatory G-ABPs that display different binding interfaces on G-actin: (i) a chimeric β-thymosin domain of Thymosin-β4 and Ciboulot proteins, which inhibits actin self-assembly by sequestering G-actin [37], (ii) profilin and (iii) a profilin-like WH2 domain of the Cordon-Bleu protein [38]. The latter two control the unidirectional assembly of actin monomers at the barbed-ends of filaments. At high, saturating concentrations of these three G-ABPs, only profilin does not interfere with G-actin-induced activation of Vn-ExoY AC activity and allows potent AC (Fig 2A). We then performed AUC-FDS experiments, monitoring the sedimentation behaviour of Alexa-488-labelled-G-actin in the presence of Vn-ExoY, profilin or both proteins. The formation of complexes of increasing size was observed by the shifts from an $s_{20,w}$ value of 3.5 S obtained for the peak of free Alexa-488-labelled actin to 4.2 S with profilin (15 kDa), 5.0 S with the larger Trx-Vn-ExoY protein (69.1 kDa), and to a higher value of 5.5 S with both proteins (Fig 2B), validating the formation of a ternary Vn-ExoY:actin:profilin complex in solution. These results are consistent with the interaction of *V. vulnificus* Vv-ExoY with actin:profilin [28]. We next investigated whether the interaction of Vn-ExoY with G-actin was modulated by the simultaneous binding of profilin. We measured the binding strength of Trx-Vn-ExoY$^{DM}$ to fluorescently labelled G-actin-ATP alone or bound to profilin using microscale thermophoresis (MST). Titration of the interaction by measuring changes in fluorescence (Fig 2C) or MST signal (S1 Fig) shows that Vn-ExoY binds with similar affinity to free or profilin-bound G-actin. The abundance of actin:profilin in eukaryotic cells [36,39] thus compensates for the modest affinity of Vn-ExoY for its likely physiological cofactor, actin:profilin.

## Assembly of actin:profilin at F-actin barbed-ends is inhibited by Vn-ExoY

To understand the effect of Vn-ExoY interaction with actin:profilin on actin self-assembly, we first examined its effect on the spontaneous assembly of actin:profilin at F-actin barbed-ends.

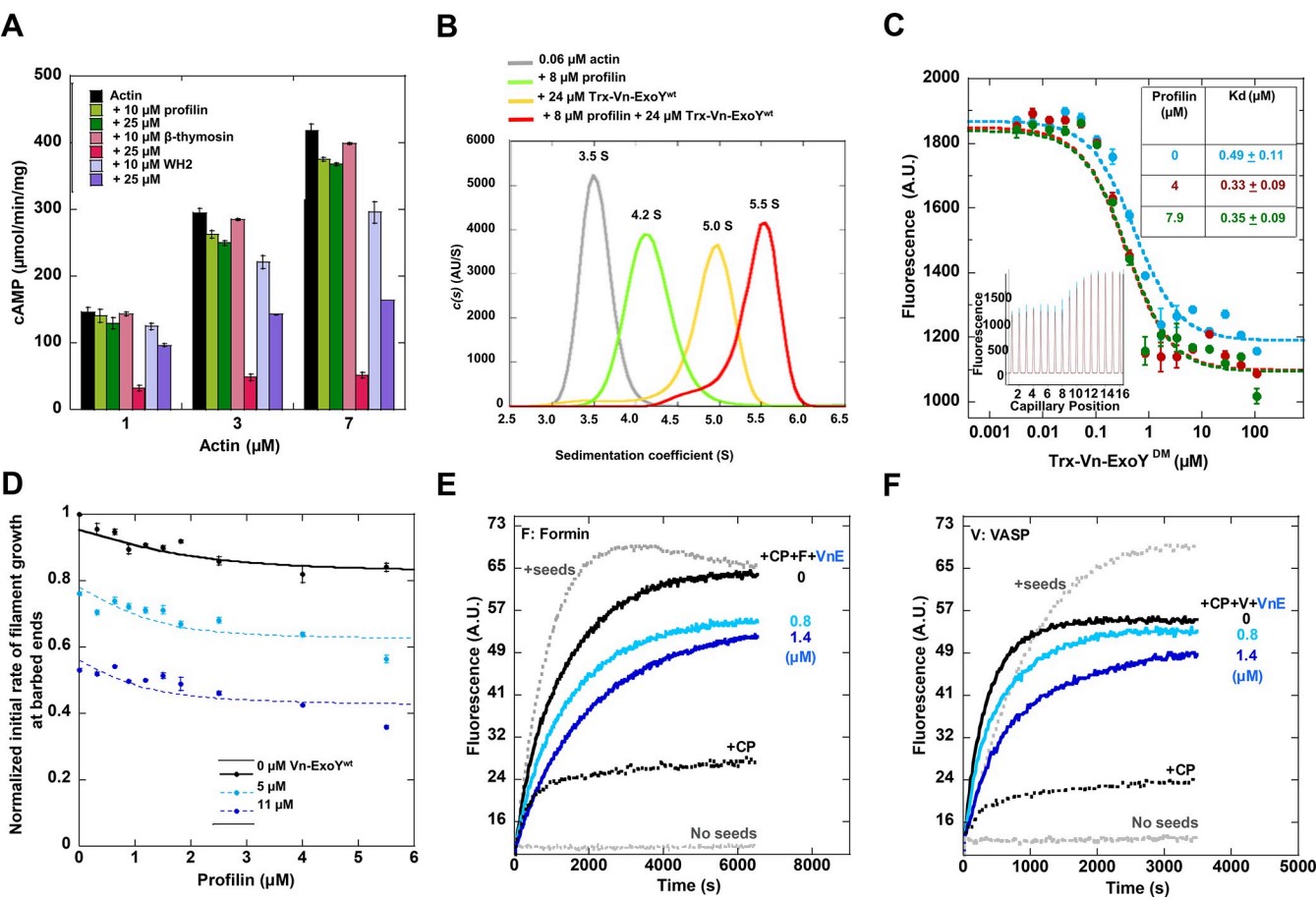

**Fig 2. Vn-ExoY is activated by profilin-bound G-actin and inhibits spontaneous or regulated assembly of actin:profilin on F-actin barbed-ends *in vitro*.**
**(A)** cAMP synthesis by G-actin-activated Vn-ExoY in the presence of G-ABPs. Reactions containing 5 ng of Vn-ExoY and Latrunculin-A-bound actin at the indicated concentrations (μM) were initiated with 2 mM ATP and incubated at 30˚C for 30 min. cAMP synthesis was measured in the absence (black) or presence of two high concentrations (10, 25 μM) of profilin (green), a sequestering β-thymosin (pink) or profilin-like WH2 (purple) domains. The Kd of their interaction with G-actin is 0.1, 0.5 and 0.5 μM, respectively [37,38]. Error bars correspond to s.d. of two independent experiments each performed in duplicate. **(B)** Formation of the Vn-ExoY:actin:profilin complex by AUC-FDS. The c(s) distribution of Alexa-488-labelled-G-actin (60 nM) was determined alone (grey line) and in the presence of either profilin (8 μM; green line), Trx-Vn-ExoY (24 μM; yellow line) or both proteins (red line). **(C)** Measurement of Trx-Vn-ExoY$^{DM}$ binding to free or profilin-bound actin from fluorescence changes in low ionic strength MST experiments. Titration of Alexa-488-labelled, Latrunculin-A-bound ATP-G-actin (0.075 μM) alone (cyan) or bound to 4 (red) and 7.9 (green) μM profilin by Trx-Vn-ExoY$^{DM}$ for a 2-fold dilution series from 108 to 0.033 μM of Trx-Vn-ExoY$^{DM}$ in MST-optimised buffer (see Methods). The inset shows the fluorescence changes in the raw data. The table shows the resulting Kd. Error bars are s.d. (n≥3). **(D)** Vn-ExoY inhibits barbed-end elongation by actin:profilin. Initial barbed-end growth rates from spectrin-actin seeds (0.36 nM) were measured as in Fig 1D using 2 μM G-actin (5% pyrenyl-labelled) and increasing concentrations of profilin in the absence and presence of Vn-ExoY at the indicated concentrations. **(E-F)** Vn-ExoY inhibits formin- (E) or VASP-mediated (F) barbed-end elongation from actin:profilin. Barbed-end growth from spectrin-actin seeds (0.13 nM), 1 μM G-actin (5% pyrenyl-labelled), and 8 μM profilin was measured in the absence or presence of 2.1 nM of capping protein (CP), 3.2 nM of FH1-FH2 domains of mDia1 formin (denoted by F) (E) or 0.22 μM of VASP (V) (F) at the indicated concentrations of Vn-ExoY (VnE).

We measured the normalised initial rate of barbed-end filament growth in the presence of increasing concentrations of profilin with or without fixed concentrations of Vn-ExoY. Without Vn-ExoY, barbed-end growth was only modestly inhibited by saturating concentrations of profilin as actin:profilin can associate with and elongate F-actin barbed-ends (Fig 2D). With Vn-ExoY, the dependence of barbed-end growth on profilin concentration shifted to lower values in a Vn-ExoY dose-dependent manner. This shift was consistent with a decrease in the G-actin pool available for barbed-end elongation due to its sequestration by Vn-ExoY. We next examined the effect of Vn-ExoY on actin:profilin assembly at F-actin barbed-ends regulated by VASP or formin processive barbed-end elongation factors. The latter bind to barbed-

ends via an F-actin binding domain (called Formin-Homology domain 2 (FH2) in formins) and favour actin:profilin recruitment to barbed-ends through their adjacent profilin-binding proline-rich motifs (PRMs, which are repeated in formin FH1 domain). Formins or VASP also counteract the barbed-end capping and growth inhibition by capping proteins (CP). The barbed-end growth of F-actin seeds was therefore monitored in the presence of both G-actin: profilin, CP and either the C-terminal FH1-FH2 domains of mouse Dia1 formin (F or mDia1-FH1-FH2, Fig 2E) or full-length human VASP (V, Fig 2F) to approach the physiological regulation of F-actin barbed-ends. When actin (1μM) is saturated with profilin (8 μM), actin nucleation is prevented, and polymerisation does not occur (horizontal dashed grey lines). Under these conditions, the addition of 0.35 nM of spectrin-actin seeds induced rapid elongation of the barbed ends of actin filaments (ascending dotted grey curves). In the presence of 2.1 nM CP, this barbed-end elongation was partially inhibited (black dashed curves), but restored by the addition of 3.2 nM mDia1-FH1-FH2 (Fig 2E, black curves) or 0.22 μM VASP (Fig 2F, black curves), as both elongation factors compete with CP at F-actin barbed ends. Further addition of Vn-ExoY at 0.8 or 1.4 μM to mDia1-FH1-FH2 or VASP in the presence of CP inhibited the barbed-end growth from actin:profilin induced by the two elongation factors (blue curves). These inhibitory effects were independent of Vn-ExoY AC activity for the duration of the kinetic experiments (S2 Fig). Thus, the interaction of Vn-ExoY with G-actin:profilin inhibits the spontaneous or VASP-/formin-mediated assembly of this complex at the most dynamic ends of F-actin.

## Crystal structures of nucleotide-free and 3'-deoxy-ATP-bound Vn-ExoY in complex with actin-ATP/ADP:profilin or actin-ATP

Next, we investigated the structural mechanism of Vn-ExoY activation by free or profilin-bound G-actin. To improve the crystallisation strategy (see Materials and Methods), we designed a chimeric protein containing three fused functional domains: Vn-ExoY extended at the C-terminus (Vn-ExoY$^{455-896}$, either wild-type or with K528M/K535I), a short profilin-binding proline-rich motif (PRM) and profilin (Fig 3A, VnE/VnE$^{DM}$-PRM-Prof). The chimera sequestered G-actin with approximately 100-fold higher affinity than the Vn-ExoY$^{412-873}$ module and exhibited strong actin-dependent AC activity at low G-actin concentrations (Fig 3B and 3C). Thus, the actin-binding regions in Vn-ExoY and profilin appear to be fully functional in this chimera. Using this chimera (wild-type or double-mutant), we solved two crystal structures at 1.7 Å resolution: (i) VnE$^{DM}$-PRM-Prof bound to actin-ADP with latrunculin B (LatB), and (ii) VnE-PRM-Prof bound to actin-ADP without LatB (Fig 3D and S2 Table). In these structures, profilin is bound to actin-ADP, the PRM to profilin, Vn-ExoY to actin, while residues 863–895 connecting the C-terminus of Vn-ExoY to the PRM are disordered (Fig 3D and S2 Table). Vn-ExoY contacts actin-ADP via its switch A and C, but only switch C is fully ordered. The Vn-ExoY switch A disorder (residues 682–727) is close to the LatB-binding site in actin. However, solving the structure without LatB only slightly reduces the switch A disorder. In the latter structure, the NBP of Vn-ExoY is further stabilised by a sulphate ion (Fig 3D, VnE-PRM-Prof-SO$_4^{2-}$:actin-ADP). On the basis of these structural data, we designed a minimal functional Vn-ExoY module (residues 455–863) by removing the C-terminal disordered residues 864–895 (Fig 3A). This Vn-ExoY module (shortened by 32 C-terminal residues) shows similar catalytic activity to our original Vn-ExoY$^{412-873}$ construct and allowed the determination of two additional crystal structures: (i) nucleotide-free Vn-ExoY bound to actin-ATP-LatB and profilin, with a sulphate ion in Vn-ExoY NBP (Fig 3E, Vn-ExoY-SO$_4^{2-}$:actin-ATP-LatB:profilin), and (ii) Vn-ExoY bound to actin-ATP-LatB with its NBP filled by a non-cyclisable ATP analogue 3'-deoxy-ATP (3'dATP, Fig 3F), at 1.7 and 2.1 Å resolution,

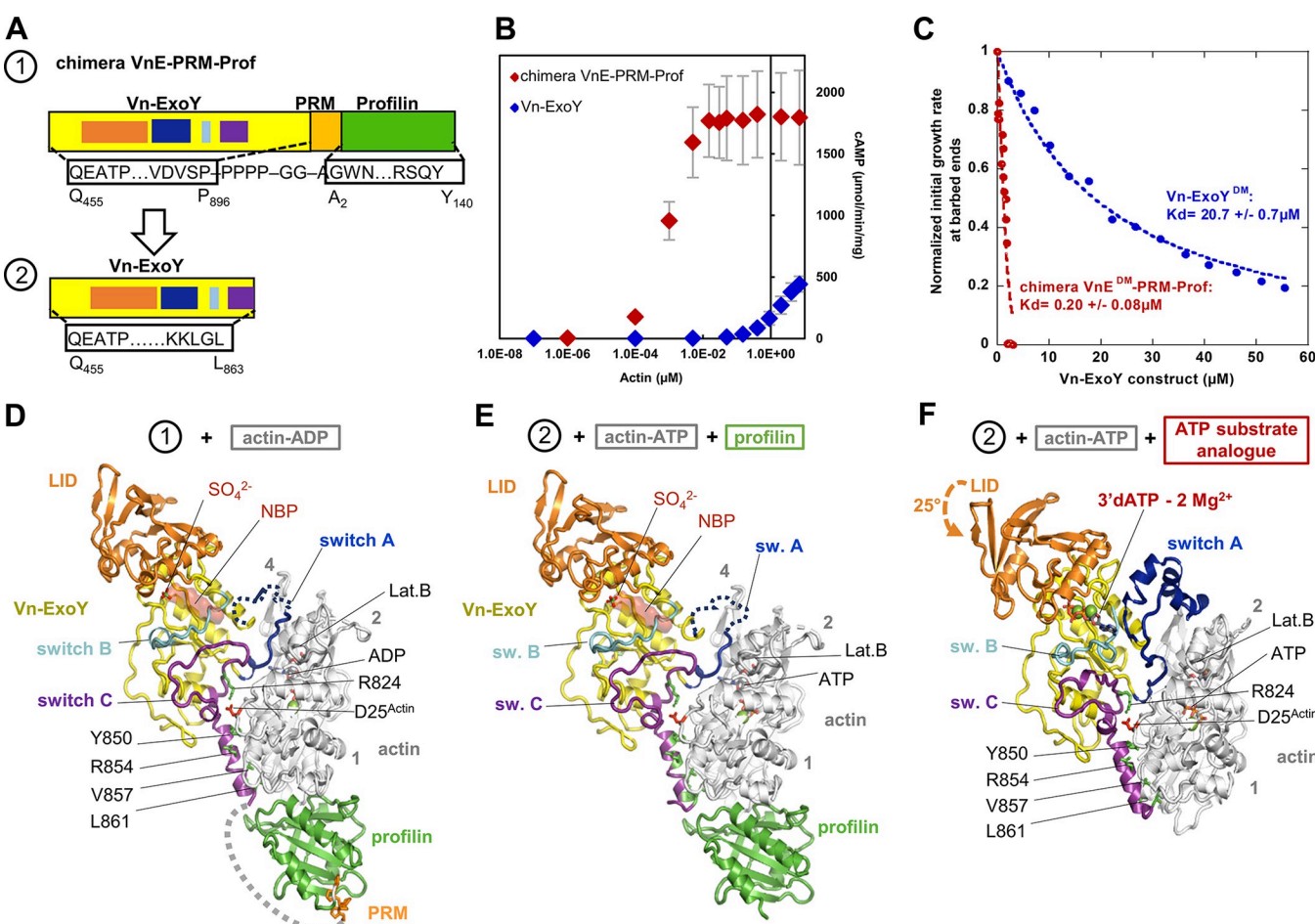

**Fig 3. Crystal structures of Vn-ExoY in different conformational states bound to actin:profilin or actin, starting with a chimeric protein containing three fused functional domains including Vn-ExoY. (A)** Domain diagrams of: (1) the VnE/VnE$^{DM}$-PRM-Prof chimera used to initiate the crystallographic study containing Vn-ExoY extended at the C-terminus (ExoY$^{455-896}$, numbering from Uniprot accession no. (AC) F0V1C5 without thousands (3000 to be added)) fused to a short PRM/Proline-Rich-Motif capable of binding to profilin, and human profilin-1 (Uniprot AC P07737), and (2) an optimised shorter Vn-ExoY$^{455-863}$ construct that remains as active as our original construct Vn-ExoY$^{412-873}$ [3]. **(B)** Synthesis of cAMP by actin-activated Vn-ExoY or VnE-PRM-Prof. VnE-PRM-Prof exhibits strong actin-dependent AC activity that saturates at much lower G-actin concentrations. Reactions containing 5 ng Vn-ExoY or 2 ng VnE-PRM-Prof and actin at the indicated concentrations were initiated with 2 mM ATP and incubated at 30°C for 10 or 30 min. Error bars correspond to s.d. of two independent experiments. **(C)** VnE$^{DM}$-PRM-Prof sequestered G-actin-ATP with 100-fold higher affinity than Vn-ExoY$^{DM}$ (residues 412–873) in the barbed-end elongation. Initial barbed-end growth rates from spectrin-actin seeds were measured using 2 μM actin (3% pyrenyl-labelled) and increasing concentrations of VnE$^{DM}$-PRM-Prof (in red) or Vn-ExoY$^{DM}$ (in blue). **(D)** Structure of the nucleotide-free VnE$^{DM}$-PRM-Prof chimera bound to actin-ADP-LatB. In Vn-ExoY, the C$_A$ subdomain and the flexible switch A, switch B, switch C and LID/C$_B$ subdomain are shown in yellow, blue, cyan, purple and orange, respectively, here and in all subsequent figures. Disordered regions are shown as dotted lines, the position of the ATP substrate in Vn-ExoY NBP by its surface in red and the four actin subdomains by numbers in grey. Conserved interactions corresponding to hotspots are indicated by black labels, with the Vn-ExoY and actin side chains shown as green and red sticks, respectively. The sulphate ion bound in the VnE-PRM-Prof-SO$_4^{2-}$:actin-ADP structure is overlaid. As shown by the multipartite interactions of our actin-bound VnE-PRM-Prof chimera, the ternary complex (Vn-ExoY:G-actin:profilin) does not prevent profilin from interacting with PRM segments (PRM in orange). **(E)** Structure of nucleotide-free Vn-ExoY in complex with actin-ATP-LatB and profilin (Vn-ExoY-SO$_4^{2-}$:actin-ATP-LatB:profilin). **(F)** Structure of 3'dATP-bound Vn-ExoY in complex with ATP-actin-LatB (Vn-ExoY-3'dATP-2Mg$^{2+}$:actin-ATP-LatB).

respectively. The Vn-ExoY-SO$_4^{2-}$:actin-ATP-LatB:profilin structure is very similar to the structures of the VnE/VnE$^{DM}$-PRM-Prof chimera bound to actin-ADP, with the Vn-ExoY switch A similarly disordered. In contrast, the Vn-ExoY-3'dATP-2Mg$^{2+}$:actin-ATP-LatB structure co-crystallised with 3'dATP shows an ordered Vn-ExoY:actin interface. This corresponds to a distinct conformational state of Vn-ExoY NBP that is more competent for catalysis, as described below.

## In the G-/F-actin-bound state, the switch-C of ExoYs ensures stable interactions while other regions remain dynamic

Vn-ExoY undergoes significant structural rearrangements at both the $C_A$ and $C_B$ subdomains between its nucleotide-free conformation bound to actin-ATP/ADP and profilin, and its 3'dATP-bound conformation in complex with actin-ATP (Fig 4A). At the Vn-ExoY:actin interface, none of the buried contacts of the Vn-ExoY switch C move, except for L827, which is relocated by the remodelling of switch A (Fig 4A and 4B). Thus, switch C appears to be essential for initiating the interaction and stabilising Vn-ExoY on actin. In the absence of 3'dATP, most of switch A is too flexible to be modelled, and the switch A portions that are ordered in the absence of ligand undergo profound rearrangements upon 3'dATP-binding (up to 10 Å on D682 and K683 Cα atoms, Fig 4A). Subsequently, the letters after the residue numbers are used to distinguish residues of different proteins or functional regions (e.g. $^{VnE-CA}$ refers to residues belonging to the Vn-ExoY subdomain $C_A$). The different conformations of Vn-ExoY bound to actin/actin:profilin with or without a nucleotide substrate analogue allow us to define the mobile "switch" regions as follows: Vn-ExoY switch A as residues Y675-V727$^{VnE-Switch-A}$ (I222-N264$^{PaE-Switch-A}$, UniProt AC Q9I1S4_PSEAE), switch B as H753-A770$^{VnE-Switch-B}$ (H291-A308$^{PaE-Switch-B}$), switch C as I819-L863$^{VnE-Switch-C}$ (K347-V378$^{PaE-Switch-C}$). Other important conformational changes near the interface occur in the parts of switch C in contact with switch A and B, and in the parts of switch B in contact with switch A and C (Fig 4A and 4B). These regions show high B-factors in all the complexes with nucleotide-free Vn-ExoY (Fig 4D–I). Near the interface, the $C_B$ subdomain of Vn-ExoY undergoes the most drastic global rearrangement with a large 25˚ rotation as a rigid body around the hinge regions 532-533$^{VnE}$ and 662-663$^{VnE}$. It thus acts as a lid (hereafter referred to as $C_B$/LID for residues K533-M662$^{VnE-LID}$, corresponding to K86-M209$^{PaE-LID}$ in Pa-ExoY) that closes on $C_A$ to stabilise 3'dATP binding (Fig 4A and 4B and S1 and S2 Movies). The conformational changes induced by 3'dATP binding bring several regions of the Vn-ExoY $C_A$ subdomain closer to actin, increasing the buried solvent-accessible area (BSA) of switch A from 338 to 707 Å$^2$, reaching the BSA value of switch C (810 and 766 Å$^2$, respectively). Vn-ExoY binds to actin over a large surface area, with switch A binding primarily to actin subdomains 2 and 4 and switch C binding to actin subdomains 1 and 3 (Fig 4C).

To validate the importance of switch C for proper positioning of ExoY on actin, we mutated conserved buried residues of switch C that remain well anchored and stable in the complexes with or without 3'dATP. Actin D25 (D25$^{Actin}$) forms a buried salt bridge at the interface core with the R824$^{VnE-switch-C}$ side chain (distances of 2.7 and 3.2 Å between N-O atom pairs, Figs 3D–3F, 4B and 4C) and was identified as a critical residue for Pa- and Vn-ExoY activation by F- and G-actin, respectively [3]. The R824A$^{VnE-switch-C}$ mutation alone results in a 130-fold reduction in Vn-ExoY G-actin stimulated AC activity (Fig 5A). The extreme C-terminus of Pa-ExoY is also critical for its binding to F-actin [40]. The corresponding region in Vn-ExoY is located at the edge of the Vn-ExoY:actin interface, where the extreme C-terminal amphipathic helix of Vn-ExoY (residues 847-863$^{VnE-switch-C}$) binds into the hydrophobic cleft between actin subdomains 1 and 3. Mutation of the most buried hydrophobic residues of this amphipathic helix to alanine results in a 240-fold reduction in its AC activity (Fig 5A). Similar interactions occur in the switch C of F-actin-bound Pa-ExoY [28]. Three basic amino-acids of Pa-ExoY near D25$^{Actin}$ may play the same role as R824$^{VnE-switch-C}$: K347$^{PaE-switch-C}$, H352$^{PaE-switch-C}$ and R364$^{PaE-switch-C}$. While the R364A mutation in Pa-ExoY switch C had almost no effect on its F-actin-stimulated GC activity, the K347A and H352A mutations caused a ~15- and ~8-fold decrease, respectively (Fig 5C), indicating that these two amino-acids together function as R824$^{VnE-switch-C}$. On the other hand, changing the three bulky hydrophobic side

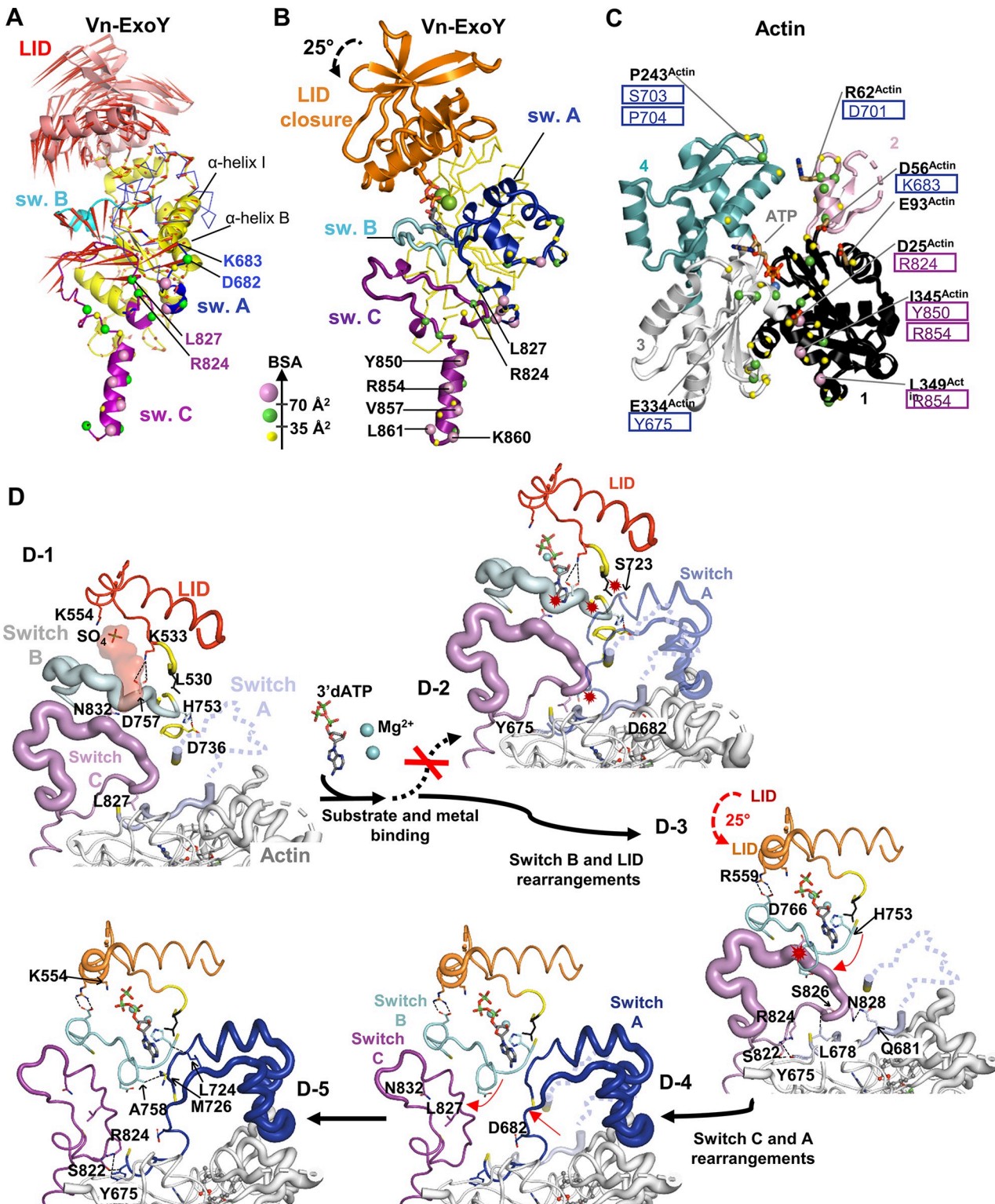

**Fig 4. Sequence diagram of the structural rearrangements in actin-bound nucleotide-free Vn-ExoY upon substrate binding. (A)** Porcupine plot illustrating the direction of Cα atom movement [85] between the nucleotide-free and 3'dATP-bound Vn-ExoY conformations reported on the Vn-ExoY conformation in Vn-ExoY-SO$_4$$^{2-}$:actin-ATP-LatB:profilin structure (actin and profilin omitted). Red arrow length proportional to Cα atom motion amplitude. Residues with significant solvent-accessible areas (BSAs) buried by actin are indicated by yellow (BSA $\leq$ 35 Å$^2$), green (between 35 and 70 Å$^2$) and pink ($\geq$ 70 Å$^2$) spheres on Cα. **(B)** The different conformation of Vn-ExoY in Vn-ExoY-3'dATP-2Mg$^{2+}$:actin-ATP-LatB structure. **(C)**

Vn-ExoY-buried actin residues on the actin structure. The four numbered actin subdomains are shown in different colours. Vn-ExoY residues are boxed with the same colour code as in (A,B). **(D)** The LID/$C_B$, switch A, B and C regions of Vn-ExoY are coloured red, light blue, grey and violet in Vn-ExoY-$SO_4^{2-}$:actin-ATP-LatB:profilin, and orange, blue, cyan and purple in Vn-ExoY-3'dATP-2$Mg^{2+}$:actin-ATP-LatB, respectively. D-1) In the nucleotide-free state, switch B and the beginning of switch C, which contacts switch A and B, are ordered but show relatively high B-factors in Vn-ExoY. The structures are drawn in a cartoon putty representation [85], where the backbone thickness is proportional to the B-factor value. The black dotted lines represent intramolecular interactions. D-2) The conformations of switch B and C, and residues (L530-F531$^{VnE}$) near the LID hinge region V532-K533$^{VnE}$ are incompatible with 3'dATP binding or the final stabilised conformation of switch A induced by 3'dATP binding. To avoid steric clashes (red explosion symbols), they have to be rearranged. D-3) Structural rearrangement of switch B and LID. LID closure is stabilised by electrostatic bonds between K535$^{VnE-LID}$ or R559$^{VnE-LID}$ and D766$^{VnE-switch-B}$ side chains. The red arrows indicate how the different regions move. D-4) Structural rearrangements of switch A and C. D-5) In this 3'dATP-bound state, switch B and C have B-factors of the same order as those in the overall structure, and the entire switch A stabilises on G-actin. Only the region of switch A (residues 686–719) that binds between actin subdomains 2 and 4 retains relatively high B-factors.

chains of the C-terminal amphipathic helix of Pa-ExoY (i.e. F367-L371-F374, Fig 5B) into alanine caused a ~12,000-fold reduction in F-actin stimulated GC activity (Fig 5C). These residues in the switch C of Vn-ExoY or Pa-ExoY are conserved in most of their counterparts in other β- and γ-proteobacteria (Fig 5D). Therefore, to transition into their catalytic conformation, these enzymes initially bind securely to their cofactor (G-/F-actin) through their switch C, while other regions remain flexible to facilitate further functional rearrangements.

## Significance of switch A and $C_B$/LID closure in stabilising nucleotide and $Mg^{2+}$

Apart from a difference in the structural topology of the two ExoY proteins (S3 Fig), the overall structure of Vn-ExoY bound to actin and 3'dATP is close to that of *V. vulnificus* Vv-ExoY bound to actin:profilin [28] (S4A Fig). In the 3.9-Å resolution cryo-EM structure, the 3'dATP ligand used to prepare the complex could not be modelled in Vv-ExoY. In both structures, switch A adopts a similar conformation between actin subdomains 2 and 4. This interaction primarily prevents Vn- and Vv-ExoY from binding along the side of F-actin. This explains their G-actin specificity, whereas the orientation of Pa-ExoY switch A bound to F-actin differs [28] (S4B Fig). Furthermore, the positioning of the Vn-ExoY switch A on G-actin prevents the association of Vn-ExoY:G-actin or Vn-ExoY:G-actin:profilin complexes with F-actin barbed-ends (S5A Fig) and is responsible for the inhibition seen in Figs 1D and 2D–2F. Switch A is also the region with the most divergent folding at the toxin:cofactor interface between G/F-actin-bound ExoYs and CaM-bound EF or CyaA (S5B and S5C Fig). Thus, this region appears to be evolutionarily essential for the cofactor specificity of NC toxins.

Our different structures reveal two major consecutive conformational changes occurring at the catalytic site of NC toxins. In the free/inactive Pa-ExoY structure [26], the A, B, and C switches were not visible as partially degraded and/or flexible. In our nucleotide-free structures of Vn-ExoY bound to G-actin-ATP/-ADP and profilin (Fig 3D and 3E), the Vn-ExoY NBP is still only partially stabilised by cofactor binding (Fig 4D–1), showing that cofactor binding is not sufficient to fully stabilise the NBP and catalytic residues of NC toxins, as previously suggested [2,24,28]. 42 to 46 of the 53 residues from switch A are still disordered. A $SO_4^{2-}$ ion is bound to the Vn-ExoY NBP via residues (K528$^{VnE-CA}$, K535$^{VnE-LID}$, S536$^{VnE-LID}$, K554$^{VnE-LID}$) that coordinate the γ- and β-phosphates of 3'dATP when this ligand is bound (Fig 4D–2). Binding of the ATP substrate and $Mg^{2+}$ ions in the Vn-ExoY NBP requires rearrangement of switch B and the supporting part of switch C (Fig 4D–3). To reach its most favourable binding position, the nucleotide must move ~4 Å (distance between the $SO_4^{2-}$ ion and 3'dATP γ-phosphate) through the LID closure (Fig 4D–3). Structural rearrangements in the region of switch C, which supports switch B, are accompanied by stabilisation of switch A and the substrate analogue (Fig 4D–4 and 4D-5). In the final stage, switch B and C are well-ordered, while the entire switch A is stabilised on G-actin (Fig 4D–5). In the Vn-ExoY-3'dATP-2$Mg^{2+}$:actin-

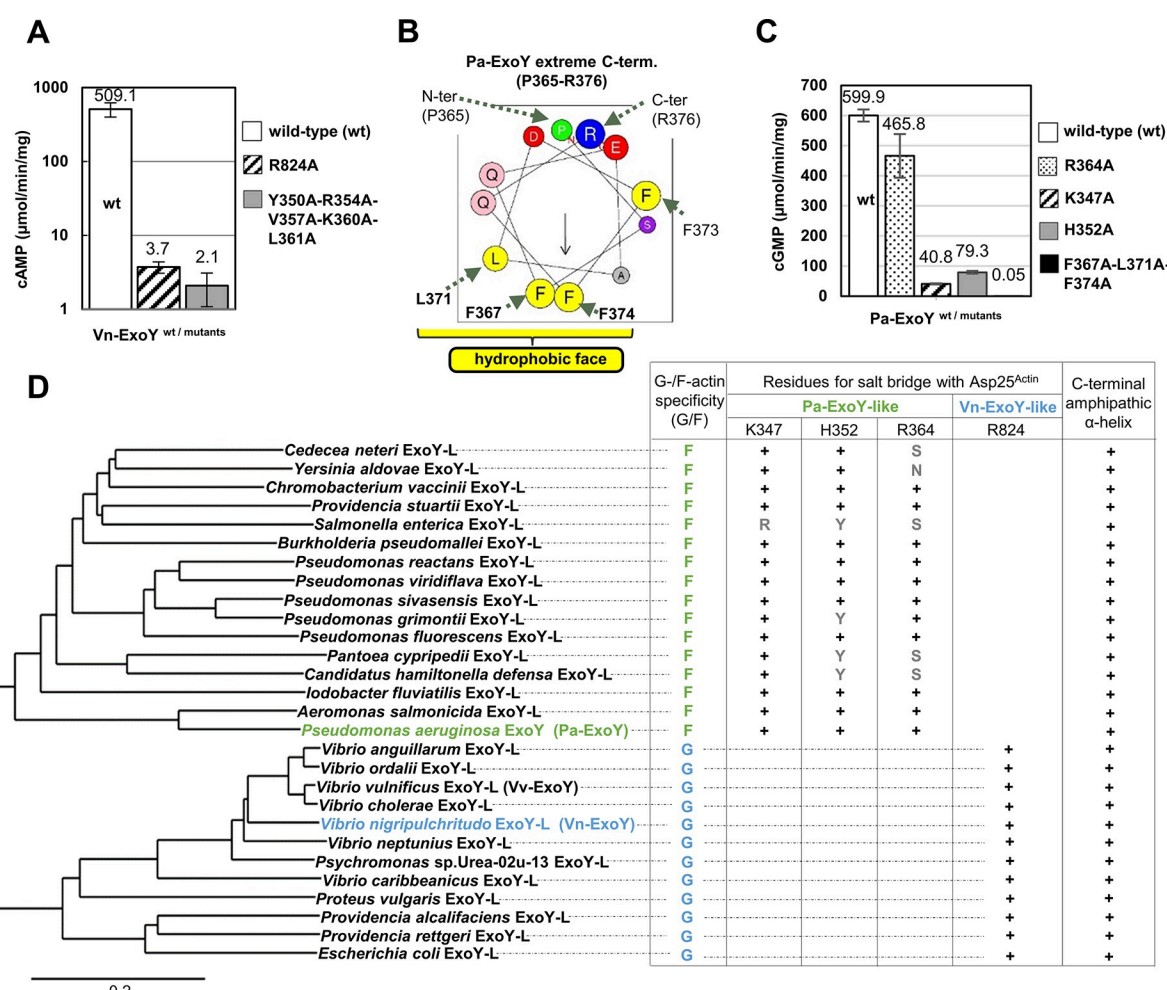

**Fig 5. Importance of the switch C at the actin:Vn-ExoY interface. (A)** cAMP synthesis catalysed by wild-type or switch C mutants of Vn-ExoY (Vn-ExoY$^{455-863}$) activated by G-actin. Reactions containing 2.5 or 5 ng for Vn-ExoY and 5, 50 or 100 ng for Vn-ExoY mutants and 3 μM G-actin-ATP-LatB were started with 2 mM ATP and incubated at 30˚C for 15 or 30 min. Values are averages of reactions with the indicated enzyme concentrations. Error bars in (A, C) correspond to s.d. of at least two replicates. **(B)** The amphipathic α-helix at the extreme C-terminus of Pa-ExoY (residues 365-376$^{PaE-switch-C}$) is shown in a helical wheel projection using the HeliQuest analysis (http:// heliquest.ipmc.cnrs.fr/). **(C)** cGMP synthesis catalysed by wild-type Pa-ExoY or switch C mutants activated by F-actin. Reactions containing 5 ng Pa-ExoY$^{wt}$ or Pa-ExoY(R364A, K347A, or H352A) or 1 μg Pa-ExoY(F367A-L371A-F374A) and 3 μM F-actin were started with 2 mM GTP and incubated at 30˚C for 10 min. In F-actin:Pa-ExoY structure [28], the K347, H352 and R364 side chains are at distances of 3.1, 4.5 and 4.1 Å from the D25$^{Actin}$ side chain, respectively. **(D)** Phylogenetic tree showing the relationships between ExoY-like proteins/ effector domains produced by different β- and γ-proteobacteria. The scale bar represents a genetic distance of 0.2 amino-acid substitutions per site. Vn- and Pa-ExoY are representative of the distant ExoYs. ExoYs are classified as Vn- or Pa-ExoY-like homologues based on both their sequence similarity and the predicted complex with actin by AlphaFold [87] (see S10 Fig for more details). A + sign indicates that the Vn- or Pa-ExoY residue/structural element is conserved at the same position in the ExoY-like homologue's sequence and predicted structure.

ATP-LatB structure, switch A ultimately contributes to the stabilisation of the nucleotide base through interactions with M726$^{VnE-switch-A}$.

## Catalytic mechanism of ExoY NC toxins activated by G- or F-actin with purine nucleotides

In the known cofactor:toxin structures, the nucleotide ligands adopt different positions and conformations in the toxin catalytic pocket, variably coordinated by one or two metal ions

(S6 and S7 Figs). The Pa-ExoY-bound 3'dGTP is coordinated by a single metal ion in the F-actin-activated Pa-ExoY structure [28]. In the structure of Vn-ExoY bound to actin and 3'dATP, the electronic density and coordination geometry suggest that 3'dATP is coordinated by two $Mg^{2+}$ ions. To unambiguously distinguish metal ions from water, we replaced $Mg^{2+}$ with $Mn^{2+}$ ions and collected an anomalous data set at the manganese absorption edge (see Materials and Methods). The anomalous difference Fourier map confirms that two $Mn^{2+}$ ions are bound to 3'dATP at the positions occupied by the $Mg^{2+}$ ions, with no significant changes elsewhere (Fig 6B). $Mg^{2+}_A$ is pentacoordinated in a square pyramidal geometry by two aspartates D665[VnE-CA] and D667[VnE-CA], and H753[VnE-switch-B], which are invariant in all NC toxins (Fig 6A), an α-phosphate oxygen of 3'dATP, and a water molecule. $Mg^{2+}_B$ is octahedrally hexa-coordinated by the same two aspartates (D665/D667[VnE-CA]), one oxygen from each of the three phosphates of 3'dATP and a water molecule.

Fig 6A summarises the main interactions observed with 3'dATP in the Vn-ExoY-3'dATP-2\*$Mg^{2+}$:actin-ATP-LatB structure, extrapolated to the natural substrate ATP. The residues of Vn-ExoY that coordinate the nucleotide and metal-ions are indicated in coloured boxes in the first inner shell. The corresponding residues in the Pa-ExoY, EF, and CyaA sequences are in the next outer shells. 9 out of the 14 substrate and $Mg^{2+}$ ion binding residues belong to regions of the enzyme that are flexible and significantly rearranged and stabilised upon binding of the cofactor, nucleotide substrate and metals. This explains the low basal activity of the free enzyme [5]. All side-chain contacts are provided by residues that are invariant or similar in Pa-ExoY, EF, and CyaA, and most interactions are conserved in CaM-activated EF bound to 3'dATP with a $Yb^{3+}$ ion and a closed LID (PDB 1k90) [24] (S7D Fig).

To assess the relevance of the observed interactions with the ATP analogue, some of the conserved residues involved were mutated in Vn- and Pa-ExoY and the effect on their AC and GC activity was investigated. H351[EF-LID] in EF has been proposed to play a role in the 3'OH deprotonation, either by accepting the 3'OH proton (general base), or by facilitating the presence of $OH^-$ ions near the ATP 3'OH group [24,27,30,32,33,41]. H351[EF-LID] is conserved in CyaA, but in ExoYs it corresponds to a lysine: K533[VnE-LID] or K86[PaE-LID] (Fig 6A). In the active site of Vn-ExoY bound to actin, 3'dATP and 2 $Mg^{2+}$ or 2 $Mn^{2+}$, the K533[VnE-LID] side chain is too distant to potentially coordinate the ATP 3'OH moiety if the latter were present. K533Q[VnE-LID] or K86Q[PaE-LID] mutations only decreased G/F-actin-activated AC and GC activity by 14- and 12-fold, respectively (Fig 6C and 6D), further suggesting that these residues are unlikely to act as a general base. Next, we examined K528[VnE-CA] and K535[VnE-LID], which are close to the bridging O3α and non-bridging O2α, O1β and O2γ oxygens of 3'dATP (Fig 6A). Both side chains are ~3.4 Å from O3α. K528[VnE-CA] is closer to O2α and O2γ (2.8–2.9 Å) than K535[VnE-LID], and K535[VnE-LID] is closer to O1β (3.3 Å). The catalytic activity of Vn- and Pa-ExoY was essentially abolished (at least ~23,000-fold reduction) by the K528Q[VnE-CA] and the corresponding K81Q[PaE-CA] mutation in Pa-ExoY. The K535Q[VnE-LID] or the equivalent K88Q[PaE-LID] mutations resulted in a 250- and 26-fold decrease in G-/F-actin-induced Vn-ExoY AC or Pa-ExoY GC activity, respectively (Fig 6C and 6D). The invariant H753[VnE-switch-B], which coordinates $Mg^{2+}_A$ and stabilises the purine base via a π-π stacking with its imidazole ring, is specific to NC toxins. $Mg^{2+}_A$ may be particularly important in the NBP of activated mammalian ACs to coordinate and polarise the 3'OH of ATP, facilitating its deprotonation as a first step in the cyclisation reaction [19,20,42]. The H753Q[VnE-switch-B] or equivalent H291Q[PaE-switch-B] mutations greatly reduced the catalytic activity of the toxins (~13,000-fold reduction, Fig 6C and 6D), suggesting a critical role for the switch-B invariant histidine and $Mg^{2+}_A$ in the catalysis of NC toxins.

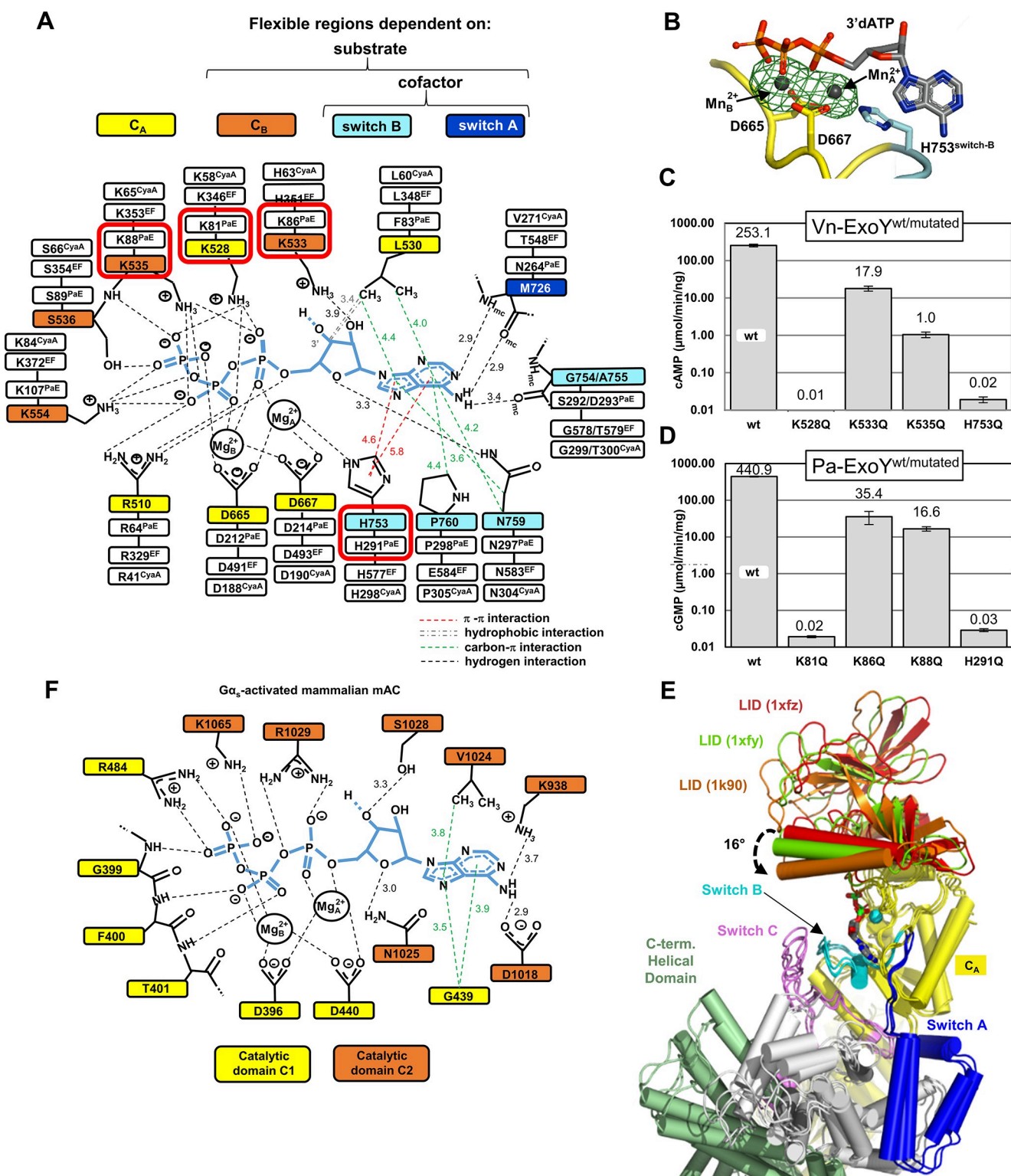

**Fig 6. Model of ATP-binding in activated Vn-ExoY and mammalian transmembrane AC (tmAC) and residues important for the purinylyl cyclase activities of Vn- and Pa-ExoY. (A)** Model of ATP-binding in activated-Vn-ExoY based on its interactions with 3'dATP in the Vn-ExoY-3'dATP-2*Mg2+: ATP-actin-LatB structure. Thresholds used for interaction detection are those of the PLIP [88] and Arpeggio [89] web servers and the PoseView tool in the ProteinsPlus web server [90]. **(B)** The model-phased anomalous difference Fourier map from data collected at 6.55 keV, shown as a green mesh contoured at 5.0 σ, confirms the presence of two Mn2+ (grey spheres) bound to 3'dATP in Vn-ExoY NBP. **(C)** cAMP synthesis catalysed by wild-type or mutants of

Vn-ExoY (Vn-ExoY[455-863]) activated by G-actin. Reactions containing 10 ng Vn-ExoY, 100 ng Vn-ExoY(K533Q) or 1 μg Vn-ExoY(K528Q, K535Q, or H753Q) and 3 μM of LatB-Mg-G-actin were started with 2 mM ATP and incubated at 30°C for 10–30 min. **(D)** cGMP synthesis catalysed by wild-type or mutants of Pa-ExoY activated by F-actin. Reactions containing 10 ng Pa-ExoY[wt]/Pa-ExoY(K86Q or K88Q) or 8 μg Pa-ExoY(K81Q or H291Q) and 3 μM of Mg-F-actin were started with 2 mM GTP and incubated at 30°C for 10–30 min. Error bars in (C, D) correspond to s.d. of at least two independent experiments. **(E)** The LID/$C_B$ subdomain of EF occupies different positions relative to the $C_A$ subdomain between the nucleotide-free (1xfz, 1xfy) [27] and nucleotide-bound (1k90) [24] structures of CaM-bound EF. It rotates between the 1xfz and 1k90 PDBs by 16° as a rigid body. The structures are superimposed on their $C_A$ subdomains shown in yellow. **(F)** Model of ATP-binding in activated class III tmAC based on the structure of the type V AC C1a/type II AC C2 heterodimer bound to the ATP analogue RP-ATPαS (PDB 1CJK) [19]. Both activated Vn-ExoY and tmAC with closed NBPs exhibit similar interactions with ATP analogues. Notably, the conserved asparagine residue N759[VnE-switch-B] or N1025[tmAC-IIC2] similarly hydrogen-bonds the ribose ring O4'-oxygen of ATP analogues, crucially stabilising the ATP ribose for proper positioning during class II and III AC catalysis. Mutations at N1025[tmAC-IIC2] lead to a 30–100 reduction in tmAC kcat [44], and mutations at N583[EF-switch-B] in EF (equivalent to N759[VnE-switch-B]) result in at least a two-order of magnitude reduction in EF catalytic activity [24].

## Structural basis for the pyrimidinyl cyclase activity of ExoYs

The pyrimidinyl cyclase activity of NC toxins has been observed *in vitro* [5,43] or in eukaryotic cells [4,16]. However, no structural characterisation has been performed yet. Vn-ExoY, EF and CyaA utilise CTP as pyrimidine nucleotide substrate, while Pa-ExoY prefers UTP but can also use CTP. To investigate how activated NC toxins can utilise small pyrimidine nucleotides as substrates, we determined the structure of Vn-ExoY bound to 3'dCTP in complex with actin-ADP at 2.2 Å resolution (Vn-ExoY-3'dCTP-2*Mn[2+]:actin-ADP-LatB). To our knowledge, this is the first structure of an activated NC enzyme in association with a pyrimidine nucleotide substrate analogue. In this structure, Vn-ExoY adopts the same overall conformation as observed in the complexes with actin-ATP, 3'dATP and two Mg[2+] or Mn[2+]. As with 3'dATP, the LID of Vn-ExoY is closed on 3'dCTP, and two metal ions coordinate the phosphate moiety of 3'dCTP. In the NBP of Vn-ExoY, 3'dCTP is stabilised by the same residues and types of interactions as 3'dATP (Fig 7A). However, the coordination of Vn-ExoY with the nucleotide ribose and base moieties is characterised by fewer and weaker hydrogen bonds, along with fewer hydrophobic interactions with 3'dCTP compared to 3'dATP. In the case of 3'dATP, the ribose experiences stabilising hydrophobic interactions with L530[VnE-CA] and is hydrogen-bonded by K533[VnE-LID] and N759[VnE-switch-B] side-chains (Fig 6A). In the activated structures of Vn-ExoY and tmAC [19], with their NBP closed on an ATP analogue, N759[VnE-switch-B] and N1025[tmAC-IIC2] similarly hydrogen-bond the O4' oxygen of the nucleotide ribose ring (Fig 6A and 6F). Such conserved stabilisation of the ATP ribose is crucial for proper ribose positioning during class II and III AC catalysis [24,44] (Fig 6F). With 3'dCTP, electrostatic interactions with the nucleotide ribose change to a single, more labile hydrogen bond between the cytidine ribose ring O4'-oxygen and a water molecule coordinated by D756[VnE-switch-B] or N759[VnE-switch-B]. The cytosine base is hydrogen-bonded by M726[VnE-switch-A] and A755[VnE-switch-B] as for the adenine base, but these residues only coordinate the N4 atom of cytosine. In contrast, with 3'dATP, several adenine atoms, namely the N1 and N6 atoms, are hydrogen-bonded through shorter hydrogen bonds. Similarly, in the Pa-ExoY:3'dGTP-1*Mg[2+]:F-actin structure, several guanine atoms are hydrogen-bonded, namely the N1, N2, and O6 atoms by the S292[PaE-switch-B], H291[PaE-switch-B], and E258[PaE-switch-A] side chains, respectively (S7C Fig). The pyrimidine base is otherwise surrounded by similar hydrophobic interactions to the purine base of 3'dATP but lacks P760 side-chain interaction with the imidazole ring of 3'dATP purine base on the solvent-accessible side. On the triphosphate moiety of 3'dCTP, the phosphates and two metal ions are tightly coordinated and less solvent-accessible than the nucleoside moiety, which is very similar to 3'dATP.

The mutations analysed above for Vn-ExoY AC or Pa-ExoY GC activity were reconsidered, and their effect on their cytidylyl or uridylyl cyclase (CC/UC) activity was examined to determine whether the conserved interactions with the nucleotide triphosphate moiety play a

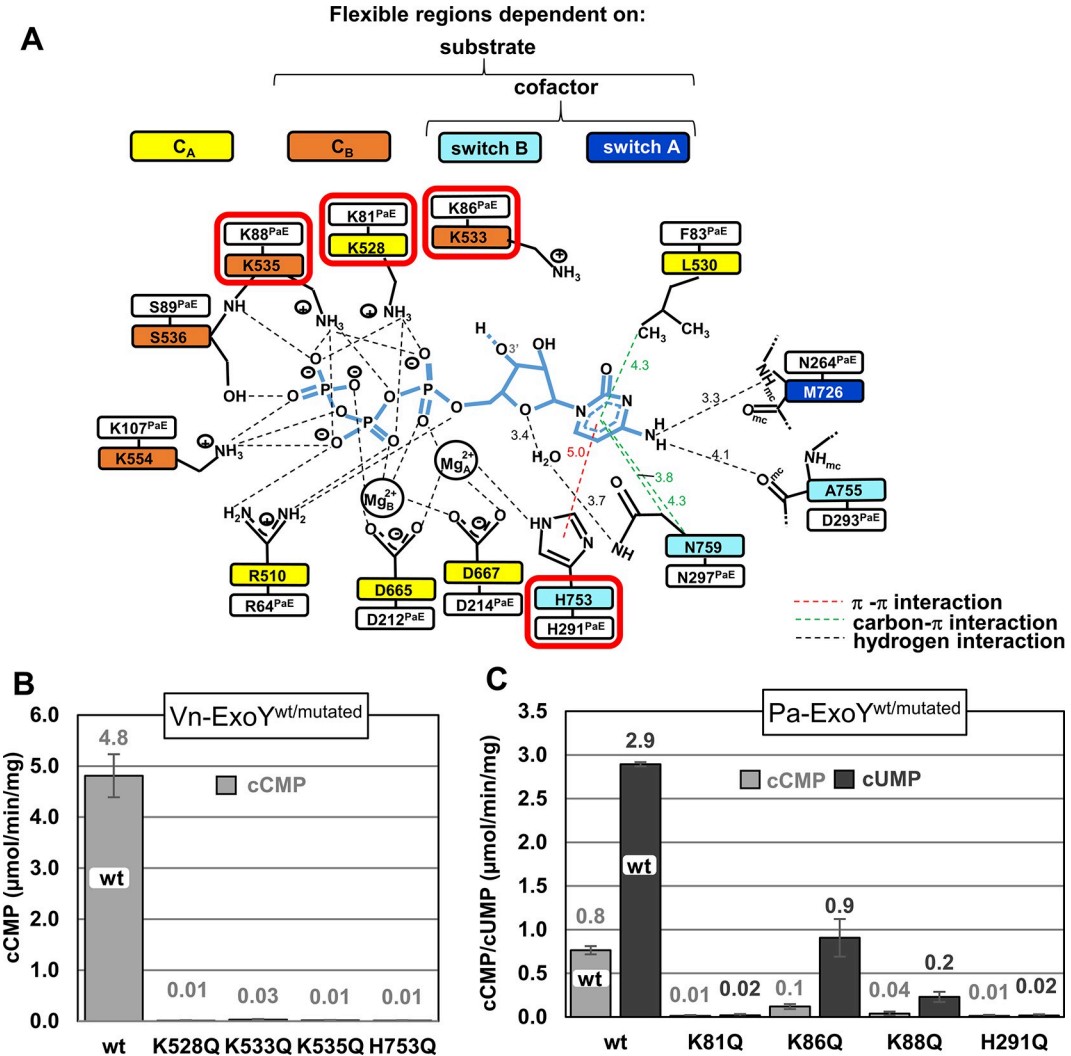

**Fig 7. Model of CTP-binding in actin-activated Vn-ExoY and residues important for the CC or UC activity of Vn- and Pa-ExoY.** (**A**) Model of CTP-binding in activated-Vn-ExoY based on its interactions with 3'dCTP in the Vn-ExoY-3'dCTP-2*Mn²⁺:ADP-actin structure. Interaction detection thresholds are as in Fig 6A and 6F. (**B**) cCMP synthesis catalysed by wild-type or mutants of Vn-ExoY activated by G-actin. Reactions containing 100 ng of Vn-ExoY^wt, or 100 or 1000 ng of Vn-ExoY (K528Q, K533Q, K535Q, or H753Q) and 6 μM of Lat.B-Mg-G-actin were started with 2 mM CTP and incubated at 30°C for 60 min. (**C**) cCMP and cUMP synthesis catalysed by wild-type or mutants of Pa-ExoY activated by F-actin. cCMP synthesis was performed by using 100 or 1000 ng of Pa-ExoY^wt or Pa-ExoY(K81Q, K86Q, K88Q, or H291Q) and 6 μM of Mg-F-actin and reactions were started with 2 mM CTP and incubated at 30°C for 60 min. cUMP synthesis was performed by using 100 or 200 ng of Pa-ExoY^wt, or 100 or 1000 ng of Pa-ExoY(K81Q, K86Q, K88Q, or H291Q) and 6 μM of Mg-F-actin and reactions were started with 2 mM UTP and incubated at 30°C for 60 min. In the absence of G-/F-actin, the basal CC and UC activity of Vn- and Pa-ExoY are 0.007 and 0.003 pmol/min/ng of the enzyme, respectively. Error bars in (B, C) correspond to s.d. of two measurements conducted at different enzyme concentrations.

similar role in the purinyl and pyrimidinyl cyclase activity of ExoYs. The mutations with the greatest effect on Vn-ExoY CC or Pa-ExoY UC activity (150- to 480-fold reduction) are those that also most affect their AC or GC activity, namely K528Q^VnE-CA or K81Q^PaE-CA and H753Q^VnE-switch-B or H291Q^PaE-switch-B, respectively (Fig 7B an 7C). As with 3'dATP, the K528^VnE-CA side chain hydrogen-bonds the α, β, and γ phosphate oxygens of 3'dCTP, and the H753^VnE-switch-B side chain stabilises both Mg²⁺_A via a hydrogen bond and the cytosine base via a π-π stacking with its imidazole ring. Together, these results demonstrate that NC toxins

use a similar two-metal-ion catalytic mechanism with purine and pyrimidine nucleotide substrates, relying on the close coordination of their triphosphate moiety for catalysis.

## Discussion

### NC toxin activation divergence and consequences of using actin:profilin as a cofactor

Bacterial pathogens utilise diverse evolutionary strategies to activate their NC toxins within host cells, recruiting either calmodulin or actin. The actin-activated ExoY-like NCs can be classified into two subgroups: Pa-ExoY-type and Vn-ExoY-type enzymes. Pa-ExoY-type ExoYs preferentially bind along filaments [3], which are abundant beneath the eukaryotic cell's plasma membrane [45] at the entry point of T3SS-delivered effectors. Our structural predictions indicate that numerous Vn-ExoY/Vv-ExoY-type orthologues (see Fig 5D) interact with actin similarly to Vn-ExoY. Except for profilin, the G-actin binding interface of Vn-ExoY overlaps and competes with that of major G-actin binding proteins (S9 Fig). This suggests that these orthologues specifically recruit G-actin in one of its most abundant cellular forms, namely the polymerisation-ready actin pool formed by profilin-bound G-actin (Fig 8, steps A-C). Notably, G-actin:profilin acts as a potent cofactor for Vn-ExoY, significantly stimulating its basal AC activity [5] by over 30,000-fold at a concentration of only 7 μM. This abundant actin:profilin pool experiences high turnover at sites of active cytoskeletal remodelling. As a countermeasure, Vn-ExoY-type toxins sequester actin:profilin, effectively prolonging their activation and cytotoxicity at their delivery sites in host cells (Fig 8, steps C-F).

The interaction of Vn-ExoY with actin:profilin is likely essential for determining subcellular localisation and activation at sites of active actin cytoskeleton remodelling. At these sites, G-actin:profilin and elongation-promoting factors play crucial roles in fine-tuning the coordination and competition of F-actin networks [46,47]. This fine-tuning of actin dynamics involves numerous multidomain ABPs and is tightly associated with cAMP signalling in vital cellular processes, including cell adhesion and T-cell migration [1,48–50]. VASP activity, regulated by the cAMP- and cGMP-dependent protein kinases (PKA and PKG) [51], is vital for immune function [52]. The recruitment of the abortive ternary complex Vn-ExoY:actin:profilin by the PRMs of multidomain ABPs (Fig 8, step F) may be of interest to pathogens to confine Vn-ExoY-like toxin activity to these discrete sites where signalling and cytoskeletal remodelling are intense and highly intertwined. These results lay the groundwork for further investigation into the subcellular compartments where G-actin:profilin-selective ExoYs generate supraphysiological levels of purine and pyrimidine cyclic nucleotides, and to elucidate their specific virulence mechanisms compared to F-actin-targeting ExoYs.

### The multistep activation of NC toxins

The identification of potential inhibitors for the ExoY NC toxins has remained elusive to date (S1 Table). We identified two critical hotspots within the switch C of Vn- and Pa-ExoY, where mutations almost completely abolish G- or F-actin-dependent NC activity (Fig 5). These sites show promise in the search for effective ExoY toxin inhibitors outside of the toxin NBP, given the stabilising and anchoring role of Vn-ExoY switch C at the Vn-ExoY:actin interface. In contrast, switch A exhibits minimal stabilisation (or only pre-stabilisation) by actin in the absence of substrate (Figs 3D–3F and 4D). On the other hand, EF and CyaA present a fully folded switch A in their nucleotide-free, CaM-bound state [24,25]. However, their $C_B$/LID subdomain can also adopt various positions in isolated CyaA [53] or in different crystal structures of CaM-bound EF (Fig 6E, as explained in more detail in S8 Fig).

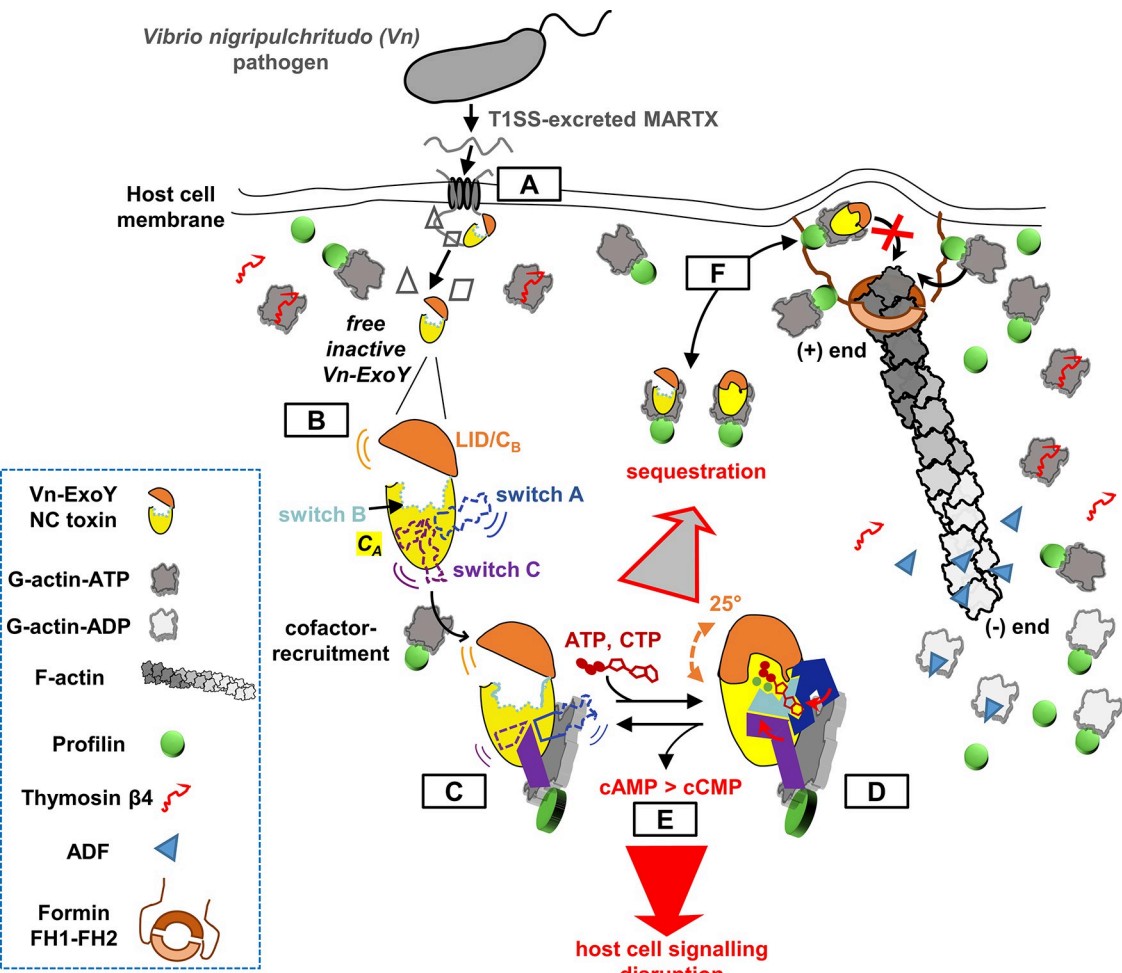

**Fig 8. Model of the activation mechanism and cytotoxic effects of Vn-ExoY elicited by its interaction with actin:profilin. (A)** Following secretion by bacterial T1SS, the large MARTX toxin binds to the target cell membrane, translocates, and releases its various MARTX effector modules into host cells [14], including Vn-ExoY. **(B)** When in isolation, the three switch regions and the CB/LID domain of Vn-ExoY are flexible and unstable, resulting in the enzyme's inactivity. **(C)** With the exception of profilin, Vn-ExoY is unable to interact with other abundant G-actin complexes that regulate actin monomers in eukaryotic cells, as its binding interface competes significantly with major cytoskeletal G-ABPs, such as the abundant regulatory Thymosin-β4 or ADF proteins (Figs 2A and S9). Vn-ExoY recruits the actin:profilin complex by docking to actin via switch C. This initial step stabilises only part of the substrate and metal ion binding sites, leading to the initiation of switch B stabilisation and bringing switch A closer to the NBP. This intermediate activated conformation allows the active site to 'breathe', particularly at the $C_B$/LID subdomain of NC toxins, enabling it to adopt a variety of positions. **(D)** Substrate and metal ion binding promotes $C_B$/LID closure of NC toxins. This conformational change enables the cofactor pre-stabilised active site of NC toxins to effectively accommodate and constrain all substrate and metal ion interactions. The large substrate-induced fit movement of the $C_B$/LID subdomain in NC toxins, achieved by simply closing and opening it, facilitates the loading, processing, and release of nucleotide substrates of varying sizes (purine/pyrimidine nucleotides). **(E)** This dynamic mechanism ensures both precise substrate specificity and efficient enzymatic turnover in NC toxins. **(F)** The G-actin:profilin complex is important for supporting filament assembly at barbed/(+)-ends. Vn-ExoY-like ExoYs sequester this complex in an abortive ternary complex for F-actin elongation. Although ABPs can recruit this ternary complex through their PRMs (present, for example, in formin as FH1), the bound G-actin cannot associate with the F-actin (+)-ends (step **F**). This unique behaviour potentially prolongs the duration of Vn-ExoY's active state during actin self-assembly dynamics. It may confine its potent NC activity, resulting in the overproduction of canonical and non-canonical cNMPs, at sites of active actin remodelling (illustrated as a membrane protrusion initiated by formin elongation factor, shown with its FH1-FH2 domains). At these sites, G-actin:profilin complexes and VASP, Arp2/3 or formin elongation-promoting factors collaboratively fine-tune the homeostasis of F-actin networks in close association with cAMP-dependent signalling [47–50].

As previously reported, cofactor binding initiates allosteric stabilisation of switch B in NC toxins, achieved by repositioning the central switch A and the C-terminal switch C towards switch B [2,24,28]. However, for ExoY NC toxins, complete stabilisation of switch A on actin necessitates substrate binding and $C_B$/LID closure. Crucially, in all cofactor-bound NC toxins, switch A plays a vital role in accurately positioning the substrate by directly coordinating the purine or pyrimidine base of nucleotides (Figs 6A, 7A and S7). Notably, both the inactive conformation of free Pa-ExoY [26] and the nucleotide-free conformations of actin:profilin-bound Vn-ExoY (Fig 3D and 3E), show the metal-binding H291$^{PaE-switch-B}$ and H753$^{VnE-switch-B}$, respectively, stabilised away from the NBP by a salt bridge with D736$^{VnE}$/D274$^{PaE}$, positioned near switch A (Fig 4D). Upon binding both the cofactor and the substrate, ExoYs exhibit a flipping motion of the H753$^{VnE-switch-B}$/H291$^{PaE-switch-B}$ towards the NBP. This invariant histidine in NC toxin switch B significantly contributes to the precise positioning of the catalytic Mg$^{2+}$$_A$ near the two invariant aspartates (D665D667$^{VnE-CA}$ or D212D214$^{PaE-CA}$) with purine or pyrimidine nucleotide substrate analogues (Figs 6A, 6B, and 7A).

## Essential elements in NC toxin purinyl and pyrimidinyl cyclase activity

Our mutational analyses in Vn-ExoY and the distantly-related Pa-ExoY revealed common elements in the NC toxin active sites that likely play a crucial role in catalysing the limiting steps in the cyclisation reaction with substrates of different sizes. Mutations of K528$^{VnE-CA}$/K81$^{PaE-CA}$ and H753$^{VnE-switch-B}$/H291$^{PaE-switch-B}$ to Gln (to preserve polar interactions with ATP or Mg$^{2+}$) significantly alter the AC/GC and CC/UC activity in Vn-/Pa-ExoY, respectively (Figs 6C, 6D, 7B and 7C). These residues are conserved in EF or CyaA, and the same mutations on the equivalent K58$^{CyaA-CA}$ and H298$^{CyaA-switch-B}$ in CyaA [54,55] or H577$^{EF-switch-B}$ in EF [24] also greatly reduce their CaM-stimulated AC activity. Our data indicate that these two residues are correctly positioned and stabilised with two metal ions only after cofactor and substrate binding and $C_B$/LID closure (Figs 6A, 7A and S7A–S7E). An essential feature allowing Vn-ExoY, and likely other NC toxins, to accommodate substrates of varying sizes in their NBP and to catalyse the cyclisation of large purine or small pyrimidine nucleotides is their remarkably similar and tight coordination with the nucleotide triphosphate moiety after closure of their $C_B$/LID subdomain. However, in the structure of Vn-ExoY bound to actin and 3'dCTP-2Mn$^{2+}$, the hydrogen bond network around the ribose and base is weaker than that observed for 3'dATP. This weaker interaction likely reduces the stability of pyrimidine nucleoside binding in the NBP of activated Vn-ExoY, which may explain why Vn-ExoY remains less efficient as CC than AC. Comparison of these structural insights into the broad NC activity of NC toxins with the functional specificities of newly identified bacterial PycC pyrimidine cyclases, which contain conserved catalytic elements of class III ACs/GCs [17], may provide intriguing insights.

## Similarities for catalysis between class II and III ACs and significance of substrate-induced fit mechanism

Our data unequivocally demonstrate that Vn-ExoY employs a two-metal-ion-assisted catalytic mechanism with purine and pyrimidine nucleotides. Given the sequence similarity and structural conservation, it is highly likely that this mechanism is shared among all bacterial NC toxins after $C_B$/LID closure (Figs 6A, 6B, 7A and S7). Interestingly, the coordination of the NBP of Vn-ExoY or of the two metal ions with the ATP-analogue bears a closer resemblance to that observed in the structure of activated class III ACs [19] rather than in the known structures of cofactor-activated NC toxins (Figs 6A, 6F, and S7 and S3 Table). The active site of transmembrane or soluble class III ACs (tmAC/sAC) is located at the interface of their two homologous

catalytic domains (C1 and C2 in tmAC, Fig 6F) and is proposed to undergo an open-closed transition during catalysis. However, in tmAC/sAC, the exact role of the open-closed transition of their dimeric NBP and the identification of the dynamic active site processes during substrate binding and turnover are yet to be fully elucidated [20]. In the closed NBP of all ACs, precise positioning of different ATP moieties is vital for catalysis. Computational studies on tmAC with a closed ATP-binding NBP [42] and on EF with an open or nearly-closed ATP-binding NBP (using PDB 1XFV/1SK6) [32] have investigated the most probable reaction sequence and key interactions during each step. The reaction model for tmAC better aligns with the numerous and tight interactions observed around 3'dATP and the two metal ions in the closed NBP of activated Vn-ExoY (S11 Fig). These interactions belong to C1 and C2 structural elements that are distant in the open NBP of tmAC [19]. Similarly, in NC toxins, NBP closure brings together ATP/CTP- and metal-ion-coordinating residues belonging to four distinct regions, which are distant from each other in the absence of cofactor and substrate. Three of these ATP/CTP- and metal-ion coordination regions are intrinsically flexible, and their positioning and stabilisation depend on both cofactor and substrate binding for switch A and B, and substrate binding for $C_B$/LID (Figs 6A and 7A). Substrate-induced $C_B$/LID closure after activation reduces the degrees of freedom of the nucleotide substrate and metal ions in the NBP (S7A–S7E Fig), thus promoting catalysis.

Class II (NC toxins) and class III ACs, despite lacking tertiary and quaternary structural homology, may share crucial determinants of catalysis. Upon activation, their distinct ATP/metal-ion coordination regions collapse towards the active site, stabilising and constraining NC interactions with the nucleotide substrate, with the metal ions, as well as substrate-metal interactions. In cofactor-bound NC toxins, the substrate-induced fit mechanism is governed by the flexible $C_B$/LID subdomain, enabling efficient enzymatic turnover through repeated $C_B$/LID closing and opening for substrate processing and product release (Fig 8, steps C-D). In contrast, in the absence of cofactor, the substrate-induced closure of the NBP prevents efficient catalysis, as switch A and B, and hence, the coordination of $Mg^{2+}_A$ and the substrate nucleoside moiety (Figs 6A and 7A), remain improperly stabilised. The initial instability and incorrect positioning of the base-coordinating switch A in actin-bound ExoY toxins suggest that substrates access the active site of NC toxins through their triphosphate moiety. This observation aligns with the ability of NC toxins to accommodate nucleotide substrates with different base sizes.

Despite the critical roles of NC enzymes in numerous signalling cascades and their relevance as therapeutic targets for human pathologies associated with dysregulation of cAMP/cGMP-signalling, high-resolution structural insights are still lacking for most of them [20,22,23,56]. Our analysis deepens the understanding of the finely-tuned functional dynamics of NC enzymes and unveils the gradual transition of the NBP of NC toxins into a catalytic conformation upon cofactor and substrate binding. These findings offer fresh structural perspectives to identify strategies for inhibiting NC toxins, which play pivotal roles in various bacterial infections [1,2,6,7].

## Materials and methods

### Protein expression and purification

For biochemical studies, unless otherwise indicated, we used the 51.6-kDa Vn-ExoY functional module from V. nigripulchritudo MARTX (Vn-ExoY[3412-3873]) corresponding to its residues 3412–3873 (numbering from Uniprot accession no. (AC) F0V1C5) [5] and shown here as residues 412–873 for the sake of clarity and simplification. Vn-ExoY[412-873] has a high sequence similarity (>85%) to all MARTX ExoY modules of *Vibrio* strains (S10 Fig). For Pa-ExoY, we

used a highly soluble maltose-binding protein (MBP) fusion of the inactive mutant Pa-ExoY[K81M] [3] in actin polymerisation assays. In enzymatic activity assays, we used Pa-ExoY constructs containing residues 20 to 378 from Unitprot AC O85345 with an N-terminal twinned-Strep-Tag (ST2). The fusion proteins of Pa-ExoY and Vn-ExoY with MBP or Trx were more suitable for dose-response analyses at high concentrations. They were more soluble than the isolated ExoY modules.

All Vn-/Pa-ExoY constructs (wild-type and mutants) were similarly expressed and purified under non-denaturing conditions using their Strep-Tag as previously described [3,5]. The construct of Vn-ExoY (wild-type) or Vn-ExoY-K528M-K535M (Vn-ExoY[DM]) with an N-terminal Thioredoxin (Trx) (designated Trx-Vn-ExoY/Trx-Vn-ExoY[DM], respectively) was customised as previously described with the N-terminal MBP fusion of Vn-ExoY [5]. The fusion constructs of Vn-ExoY and Pa-ExoY (Pa-ExoY residues 20 to 378 from Unitprot AC O85345) with an N-terminal twinned-Strep-Tag (ST2) or Thioredoxin (Trx, 11.6 kDa), designed as follows: ST2-(PreScission-cleavage-site)-(Vn-/Pa-ExoY) (denoted Vn-/Pa-ExoY in the text and cloned by Genscript) or HisTag-Trx-(PreScission-cleavage-site)-(Vn-ExoY)-ST2 (denoted Trx-Vn-ExoY) were cloned into pET29b vector using BamHI and XhoI restriction sites [5]. Expression was carried out in BL21 (DE3) bacteria grown in Luria-Bertani (LB) at 37°C with shaking until an optical density at 600 nm of 0.6 was reached. After a heat shock on ice up to 16°C, expression was induced by 1 mM β-isopropylthio-D-galactoside (IPTG) for 15 h (overnight) at the post-heat shock temperature, i.e. 16°C. Cells were washed in PBS buffer, centrifuged, frozen in liquid nitrogen and stored at -20°C until purification. Purified Vn-ExoY constructs were stored in 15 mM Tris pH 8.0, 100 mM KCl, 2 mM $MgCl_2$, 1 mM DTT and purified Pa-ExoY constructs were stored in 20 mM Tris pH 8.5, 400 mM NaCl, 5 mM $MgCl_2$, 1 mM DTT. The ST2- or Trx-fusion constructs of Vn-ExoY and Pa-ExoY showed similar specific activities for cAMP and cGMP synthesis, respectively, as the PreScission protease-cleaved Vn- or Pa-ExoY constructs. The purity of the wild-type and mutant Vn-/Pa-ExoY proteins was assessed on Coomassie Blue-stained SDS/PAGE gels (S12 Fig). The proteins have identical circular dichroism spectra and elute as monomeric proteins on calibrated size-exclusion chromatography columns (S13 Fig).

α-actin (UniProt P68135) was purified from rabbit skeletal muscle as previously described [3,57] and stored in G-buffer (5 mM Tris-HCl, pH 7.8, 1 mM DTT, 0.2 mM ATP, 0.1 mM $CaCl_2$, and 0.01% $NaN_3$). Actin was pyrenyl-labelled at cysteine 374 [58]. Human profilin [59], CP [60], mDia1 FH1-FH2 [61], human VASP [62], Thymosin β4 [37], WH2 domain, and full-length human gelsolin [59] were used. Spectrin-actin seeds were purified from human erythrocytes as previously described [63]. ADP-actin was prepared from Ca-ATP-actin and converted to Mg-ADP-actin as previously described [3]. Latrunculin A was purchased from tebu-bio (produced by Focus Biomolecules). A 2-fold excess was added to G-actin and incubated for 10 min at room temperature before use in the activity assays.

## Pyrene-actin polymerisation assays

Actin polymerisation was monitored at 25°C by the increase in fluorescence of 2–10% pyrenyl-labelled actin (λexc = 340 nm, λem = 407 nm) using a Safas Xenius model FLX spectrophotometer (Safas, Monaco) with a multiple sampler device. Actin-Ca-ATP was converted to G-actin-Mg-ATP by the addition of 1/100 (vol./vol.) of 2 mM $MgCl_2$ and 20 mM EGTA just prior to the experiments. Spontaneous ATP-G-actin polymerisation assays contained 4 μM G-actin (5% pyrene-labelled) and were performed in a final buffer containing 50 mM KCl, 2 mM $MgCl_2$, 6 mM Tris-HCl pH 8, 1 mM TCEP, 1.5 mM ATP, 0.5 mM cAMP and additionally 0.5 mM GTP for Pa-ExoY experiments. Spontaneous ADP-G-actin polymerisation reactions

containing 10 μM G-actin (10% pyrene-labelled) were performed in a final buffer containing 30 mM KCl, 2 mM MgCl₂, 5 mM Tris-HCl pH 7.8, 0.5mM cAMP, 3mM TCEP. Seed polymerisation reactions contained 1–2 μM actin (3–10% pyrene-labelled), spectrin- or gelsolin-seeds, profilin, CP, VASP, formin mdia1 as indicated and were performed in a buffer containing 15 mM Tris pH 7.8 (or HEPES pH 7.5), 50–100 mM KCl, 2–4.5 mM MgCl2, 1–1.5 mM ATP, 1 mM cAMP and 2 mM TCEP. Polymerisation assays with mDia1 FH1-FH2 formin (Fig 2E) and human VASP (Fig 2F) were performed using 1 μM G-actin (5% pyrene-labelled), profilin (8 μM), spectrin-seeds (0.13nM) in a final buffer containing 50 mM KCl, 4.7 mM MgCl₂, 15 mM HEPES pH 7.5, 2.3 mM TCEP and 1.5 mM ATP. In experiments with VASP or formin in the presence of CP (2.4 nM), VASP (0.22 μM) or formin (3.2 nM), proteins were incubated in the reaction mixture for 2 min before the addition of CP, which was then incubated for a further 2 min. After the addition of CP, Vn-ExoY was added and fluorescence was recorded after 2 min incubation. The actin polymerisation kinetics presented in Figs 1A–1C, 2E, 2F, and S2 correspond to single polymerisation kinetics. They are representative of three independent experiments.

The initial rates of filament growth from the barbed or the pointed ends were measured using spectrin-actin seeds for barbed end elongation or gelsolin-actin seeds for pointed end elongation. They were determined from linear fits to data collected in triplicate during the first 2–3 min of elongation. These reactions contained 1.5 μM G-actin (5% pyrene-labelled) and were performed in a final buffer containing 28 mM KCl, 2 mM MgCl₂, 10 mM Tris-HCl pH 7.5, 2 mM ATP, 1 mM DTT.

The equations describing the G-actin sequestration and profilin-like activities of an actin-binding protein (ABP) based on the concentration dependence of rates of barbed and pointed end growth have been previously documented [37,64]. In summary, purely sequestering actin-binding proteins (C) form a CA complex with G-actin (A) that do not participate in either barbed or pointed end growth. The following equation describes the changes in the initial growth rate, denoted v.

$$v/v_0 = ([A0]-[CA]-[Cc])/([A0]-[Cc]) \qquad (Eq\ 1)$$

$$[CA] = ([C0] + [A0] + K_D \pm (([C0] + [A0] + K_D)^2 - 4[A0][C0])^{1/2})/2 \qquad (Eq\ 2)$$

where [Cc] is the actin critical concentration, [A0] is the total concentration of G-actin in the growth assay and [CA] is the concentration of complex, [C0] is the total concentration of the actin-binding protein (C corresponding here to Vn-ExoY-containing constructs), and $K_D$ is the equilibrium dissociation constant for the binding between the actin-binding protein (C) and G-actin (A). V0 is the growth rate measured in the absence of protein C (V0 = $k_+$[F]([A0]-Cc), where $k_+$ is the rate constant for G-actin association with barbed-ends and [F] is the concentration of F-actin seeds).

$K_D$ was derived by fitting the dependence of the growth in Eq 1 on C concentration. Vn-ExoY leads to complete inhibition of pointed and barbed end growth by binding to G-actin with similar affinity in barbed and pointed end growth assays (Fig 1D).

The effect of the chimera VnE^DM-PRM-Prof on the initial rate of barbed end elongation of pre-assembled actin-spectrin seeds (0.3 nM) was determined by a linear fit to the first 150 seconds of the reaction performed by using 2 μM G-actin (3% pyrene-labelled) in a final buffer containing 50 mM KCl, 2 mM MgCl₂, 50 mM Tris-HCl pH 7.8, 0.5 mM cAMP, 1 mM ATP and 2 mM TCEP (Fig 3C).

The effect of Vn-ExoY on the initial rate of barbed end elongation of pre-assembled actin-spectrin seeds (0.36 nM) with increasing concentrations of profilin in the absence and presence of Vn-ExoY$^{wt}$ at the indicated concentrations was determined by a linear fit to the first 150 s of assembly (Fig 2D). These reactions were performed using 2 μM G-actin (5% pyrene-labelled) in a final buffer containing 100 mM KCl, 2 mM MgCl$_2$, 15 mM Tris-HCl pH 7.8, 1 mM cAMP, 1 mM ATP, and 2mM TCEP. Elongation rates were normalised to the barbed-end elongation rate in the absence of both profilin and Vn-ExoY. The plot of the effect of profilin on barbed end elongation was fitted using the equation:

$$v/v_0 = 1 - v_{min}/v_0 * Cc/(A_0 - Cc) - (P_0 + A_0$$
$$+ K_D - ((P_0 + A_0 + K_D)^{\wedge}2 - 4*A_0*P_0)^{\wedge 0.5})/(2*(A_0 - Cc))*(1 - V_{min}/v_O) \qquad \text{(Eq 3)}$$

where v is initial elongation rate measured at profilin concentration [P], $V_0$ initial elongation rate in the absence of profilin ($V_0 = k_+[F]([A_0]-Cc)$, where $k_+$ is the rate constant for G-actin association with barbed-ends and [F] is F-actin seed concentration), $V_{min}$ initial elongation rate in the presence of saturating amounts of profilin, Cc critical concentration of barbed ends, $A_0$ total G-actin concentration, $P_0$ total profilin concentration and $K_D$ profilin:G-actin equilibrium dissociation constant. In the presence of Vn-ExoY, good fits to the data were obtained by considering in (Eq 3) that Vn-ExoY initially sequesters G-actin (preventing its assembly onto F-actin barbed ends) with or without profilin with a Kd of ~12–14 μM as determined in Fig 1D–1E. This led to the use of the following equations:

-with 5 μM Vn-ExoY:

$$v/v_0 = 0.8 - V_{min}/v_0 * Cc/(A_0 - 0.51 - Cc) - (P_0 + A_0 - 0.51$$
$$+ K_D - ((P_0 + A_0 - 0.51 + K_D)^{\wedge}2 - 4*(A_0 - 0.51)*P_0)^{\wedge}0.5)/(2*(A_0 - 0.51 - Cc))*(0.8 - v_{min}/v_o).$$

-with 11 μM Vn-ExoY:

$$v/v_0 = 0.6 - v_{min}/v_0 * Cc/(A_0 - 0.88 - Cc) - (P_0 + A_0 - 0.88 + K_D - ((P_0 + A_0 - 0.88 + K_D)$$
$$\wedge 2 - 4*(A_0 - 0.88)^* P_0)^{\wedge}0.5)/(2*(A_0 - 0.88 - Cc))*(0.6 - v_{min}/v_0).$$

For Figs 1D, 2D, and 3C, initial rate measurements were performed in triplicate. Results are representative of at least two independent experiments.

## Analytical ultracentrifugation

Fluorescence-detected sedimentation velocity (FD-SV) experiments were performed using a Beckman XL-I (Beckman-Coulter, Palo Alto, USA) analytical ultracentrifuge (AUC) with an An-50Ti rotor equipped with an Aviv fluorescence detection system (AU-FDS, Aviv Biomedical). Actin was fluorescently labelled with Alexa-488 N-hydroxysuccinimide (NHS) ester (A20000, Thermo Fischer Scientific) as previously described [5].

Trx-Vn-ExoY, which resulted in higher purification yields and protein solubility at high concentrations, was preferred for titration FD-SV experiments. The buffer used for AUC experiments was (53 or 100 mM KCl, 15 mM Tris pH 7.5, 3 mM ATP, 4 mM CaCl$_2$, 2 mM MgCl$_2$, 0.5 mM Na-pyrophosphate, 2 μM latrunculin A, 1% glycerol, 0.25 mg/mL BSA). A similar sedimentation coefficient ($S_{20,w}$) was obtained for G-actin in both buffers. Samples and buffers were centrifuged at 15,000 g for 10 min prior to AUC experiments and solvent property measurements. Alexa-488 labelled G-actin at a final concentration of 60 nM (0.632 μM total G-actin, 9.5% Alexa-488-labelled) was loaded into two-sector 12 mm path-length Epon charcoal-filled cells. 400 μL of samples were centrifuged at 40,000 rpm (116,369 g) at 20˚C. Fluorescence data were collected at an excitation wavelength of 488 nm and emission wavelengths

between 505 and 565 nm, with all cells scanned simultaneously every 6 min. Sedimentation velocity data were analysed using SEDFIT software [65] and the partial specific volume used for Alexa-488-G-actin was 0.723 mL/g. Buffer viscosities and densities were measured experimentally with an Anton Paar microviscosimeter and density meter (DMA 4500). Sedimentation velocity distributions were superimposed using GUSSI software [66] and peaks were integrated to obtain weighted-average binding isotherms (sw). The generated binding isotherms were then fitted using Sedphat software [67]. The c(s) distribution shown in Figs 1E and 2B represents a single experiment representative of at least two independent experiments.

## Quantification of cAMP or cGMP synthesis in vitro

Vn- and Pa-ExoY-catalysed synthesis of cAMP, cGMP, cCMP or cUMP was measured in 50 μL reactions, as described previously [3,5], containing 50 mM Tris pH 8.0, 0.5 mg/mL BSA, 200 mM NaCl, 1 mM DTT, $MgCl_2$, 2 mM NTP (NTP corresponding to either ATP, GTP, CTP, or UTP) spiked with 0.1 μCi of [α-$^{33}$P] NTP, Vn- and Pa-ExoY and indicated amounts of Mg-Actin-ATP.

## Microscale Thermophoresis experiments

Binding assays were performed in MST-optimised buffer equivalent to a low-ionic, non-polymerising G-buffer (5 mM Tris, pH 7.8, 0.1 mM $CaCl_2$, 0.2 mM ATP) with 0.05% Tween-20 on a Monolith NT.115 Microscale Thermophoresis (MST) device using standard treated capillaries (NanoTemper Technologies). Actin (final labelled-actin concentrations adjusted to 60 nM) was titrated from 108 to 0.033 μM by a 16-step 2-fold dilution series of Trx-Vn-ExoY$^{DM}$. All binding experiments were repeated three times.

## Chimeric protein design containing Vn-ExoY, a short proline-rich motif (PRM), and profilin (VnE-PRM-Prof) for structural studies

We failed to crystallise our first functional Vn-ExoY$^{412-873}$ construct [5] with ATP/ADP-G-actin, possibly because our functional MARTX Vn-ExoY fragment contained too many flexible regions to crystallise. To improve the crystallisation strategy, we designed a multi-modular chimeric protein containing C-terminally extended Vn-ExoY (residues 455 to 896), a short PRM/proline-rich motif capable of binding to profilin (consisting of five consecutive prolines with the C-terminal Pro896 of Vn-ExoY and four additional prolines, and two glycines as a very short linker/spacer to connect the PRM to the N-terminus of profilin) and profilin (VnE-PRM-Prof in Fig 3A). The chimera VnE-PRM-Prof with an N-terminal twinned-Strep-Tag (ST2) and PreScission-cleavage-site (ST2-(PreScission-cleavage-site)-VnE-PRM-Prof) was cloned by Genscript. The VnE-PRM-Prof chimera was designed to: i) more effectively prevent actin self-assembly during crystallisation by sequestering G-actin simultaneously through Vn-ExoY and profilin; and ii) capture a functional four-component complex, some of which interact with only modest affinities. The chimeras were designed before the recent availability of the AlphaFold protein-prediction tool [68] or the recent cryo-EM structures of F-/G-actin-bound ExoY homologues [28]. Design was based on Pa-ExoY structure [26], ab initio models of Vn-ExoY, and protein-protein docking experiments with ab initio models of Vn-ExoY or Pa-ExoY and experimental distance constraints using FRODOCK software [69]. The distance constraints were based on the known contact regions between actin and Vn-ExoY or Pa-ExoY [3,5] or structural similarities with CaM-activated EF and CyaA AC domains (e.g. to impose the proximity of switch A and C in the protein-protein interface) [24,25]. Docking solutions needed to be filtered by experimental distance constraints to select a minimum number of equally likely solutions. We chose a long Vn-ExoY/profilin linker to accommodate different

plausible docking solutions. We included a PRM to stabilise the Vn-ExoY C-terminus to profilin N-terminus junction and to study profilin-actin recruitment by PRMs of ABPs like formins or VASP. The chimera, named VnE-PRM-Prof, thus contains three main functional domains: Vn-ExoYwt/DM (residues Q455 to P896), a short PRM of 4 prolines (leading with Vn-ExoY P389 to the following sequence PPPPP (or P5), which allows profilin binding [70]) and human Profilin I sequence (Fig 3A). The experiments shown in Fig 3B and 3C confirmed the suitability of the chimera was for the analysis of Vn-ExoY interaction and activation by profilin:actin and for the crystallization of stable complexes. Using the VnE-PRM-Prof$^{wt/DM}$ chimeras, we obtained the first high-resolution diffracting crystals of functional complexes with actin-ADP and further optimised and shortened the Vn-ExoY boundaries (Fig 3A, residues 455–863) for subsequent structural studies. The chimera contains both Vn-ExoY and profilin, which can bind to G-actin simultaneously without disturbance or competition between each other. Consequently, the binding of both Vn-ExoY and profilin to G-actin synergizes, leading to a 100-fold increase in the affinity for G-actin compared to the isolated, wild-type Vn-ExoY (Fig 3C).

### Protein crystallization

Complex 1 [Vn-ExoY(residue Q455 to P896 with K528M, K535I) fused to PRM-profilin:ADP-actin-LatB] of the chimera VnE$^{DM}$-PRM-Prof [i.e. Vn-ExoY(residue Q455 to P896 with K528M, K535I) fused to PRM-profilin] with ADP-Mg$^{2+}$-actin-latrunculin-B was concentrated to 21 mg mL$^{-1}$ in the presence of 4 mM ADP, 11 mM MgCl$_2$, 22 mM KCl, 0.3 mM latrunculin B in DMSO (<2% v/v), 9 mM HEPES pH 6.8, 1.8 mM TCEP, 1 mM DTT, 0.008% NaN$_3$, then mixed with 30% PEG4000, 0.2 M Li$_2$SO$_4$, 0.1 M Tris-HCl pH 8.5 in a 1:1 v/v hanging drop. Crystals were soaked in the mother liquor supplemented with an increased amount of PEG4000 (35%) before flash freezing in liquid nitrogen. Complex 2 [SO$_4$$^{2−}$-bound Vn-ExoY (residue Q455 to P896) fused to PRM-profilin:ADP-actin] of the chimera VnE$^{wt}$-PRM-Prof [i.e. Vn-ExoY(residue Q455 to P896 fused to PRM-profilin] with ADP-Mg$^{2+}$-actin was concentrated to 12 mg mL$^{-1}$ in the presence of 20 mM cAMP, 4 mM ADP, 8 mM MgCl$_2$, 40 mM KCl, 40 mM NaCl, 10 HEPES pH 7.5, 1.6 mM TCEP, 0.008% NaN$_3$, and mixed with 30% PEG4000, 0.2 M Li$_2$SO$_4$, 0.1 M TrisHCl pH 8.5 in a 2:1 v/v hanging drop after microseeding. Crystals were soaked in the mother liquor supplemented with freshly prepared cAMP solution (5 mM) and an increased amount of PEG4000 (35%) for cryoprotection. No cAMP binding to Vn-ExoY was observed in the electron density. Ternary complex 3 [SO$_4$$^{2−}$-bound Vn-ExoY$^{wt}$(residue Q455 to L863):ATP-actin-LatB:profilin] was concentrated to 20 mg mL$^{-1}$ in the presence of 20 mM cAMP, 0.2 mM ATP, 20 mM MgCl$_2$, 0.25 mM latrunculin B, 0.08 M (NH$_4$)$_2$SO$_4$, 0.08 M LiCl, 14 mM MES pH 6.5, 1.6 mM TCEP and mixed with 29% PEG3000, 0.3 M Li$_2$SO$_4$, 3% dioxane, 0.1 M TrisHCl pH 8.5 in a 1:1 v/v hanging drop after microseeding. Crystals were soaked in the mother liquor supplemented with 5% v/v PEG400, freshly prepared cAMP solution (5 mM), and an increased amount of PEG3000 (33%) for cryoprotection. Binary complex 4 [(3'dATP-2Mg$^{2+}$)-bound Vn-ExoY$^{wt}$(residue Q455 to L863):ATP-actin-LatB] was concentrated to 10 mg mL$^{-1}$ in the presence of 15 mM 3'dATP, 0.34 mM ATP, 15 mM MgCl$_2$, 0.15 mM latrunculin B, 13 mM KCl, 86 mM LiCl, 7 mM HEPES pH 8.5, 1.2 mM TCEP and mixed with 26% PEG3350, 26% glycerol, 0.03 M Li$_2$SO$_4$, 0.1 M Tris pH 8.5 in a 2:1 v/v hanging drop after microseeding. For complex 5, complex 4 crystals were soaked in the former reservoir solution supplemented with 0.01% NaN$_3$ and 15 mM MnCl$_2$ for varying periods up to 4 hours. Crystals collapsed after this time. By anomalous diffraction of Mn, we confirmed the presence of two metal ions in the active site of Vn-ExoY in complex 5 [(3'dATP-2Mn$^{2+}$)-bound-Vn-ExoY$^{wt}$:ATP-actin-LatB]. We have previously shown that Vn-ExoY exhibits similar NC activity in the presence of Mg$^{2+}$ or Mn$^{2+}$ ions [5]. Binary complex 6 [(3'dCTP-

$2Mn^{2+}$)-bound-Vn-ExoY$^{wt}$(residue Q455 to L863):ADP-actin-Lat.B] was concentrated to 10 mg mL$^{-1}$ in the presence of 15.2 mM 3'dCTP, 1.6 mM ADP, 20 mM MgCl$_2$, 0.2 mM latrunculin B, 23 mM KCl, 70 mM LiCl, 8 mM HEPES pH 8.5, 4 mM TCEP and mixed with 17% PEG4000, 17% glycerol, 0.01 M Li$_2$SO$_4$, 0.1 M Tris pH 8.5, 5 mM MgCl$_2$, 15 mM MnCl$_2$, 1% 1-butyl-2,3-dimethylimidazolium tetrafluoroborate (Ionic Liquid 18 from the Ionic Liquid Screen (Hampton Research)) in a 1:1.2 v/v hanging drop. After crystal growth, the crystals were soaked in the mother liquor supplemented with 10 mM ADP for over one month to saturate actin with ADP. Crystals were soaked in the mother liquor supplemented with 6 mM ADP and an increased amount of both PEG4000 (22%) and glycerol (22%) for cryoprotection.

Prior to the crystallisation of all protein complexes, Prescission protease was added at the end of preparation of the complexes (using 0.5 unit of Prescission protease from Cytiva Co. per 100 µg of proteins). The mixtures were then incubated at 5˚C for 16 h to cleave the ST2 prior to crystallisation.

## Data collection and processing

The data were collected at 100 K on the PROXIMA-1 and PROXIMA-2 protein crystallography beamlines at the SOLEIL synchrotron (Saint Aubin, Université Paris-Saclay, France) using the *MXCuBE* application [71] and checked on the fly using the *XDSME* interface [72]. The X-ray energy was set to the Mn *K*-edge energy of 6.64 keV for the complex 5 (12.6 keV was used for the other crystals). All data sets used for structure solution and refinement were reprocessed using *autoPROC* [73] running *XDS* [74] for data indexing and integration, *POINTLESS* [75], and *AIMLESS* [76] for data reduction scaling and calculation of structure-factor amplitude and intensity statistics. For complexes 4 and 5, respective data sets from four and three different isomorphous crystals respectively were merged. Finally, *STARANISO* [77] was used to process and scale the anisotropy of the diffraction data, assuming a local I/σ(I) of 1.2. Thus, with the exception of crystal 3, the diffraction was extended to higher angles in the **a**$^*$/**c**$^*$ plane, suggesting the inclusion of useable data beyond the spherical resolution.

## Structure determination, refinement and analysis

The structures were solved by the molecular-replacement method with Phaser [78], using the structure of free Pa-ExoY [26] and the profilin-actin structure (PDB 2pav) [79] as search models to place one (complexes 1 to 3) or two copies (complexes 4 and 5) per asymmetric unit. The first initial model was automatically built using Buccaneer [80] and completed using Coot [81]. Refinement, leaving 5% of the reflections for cross-validation, was performed with BUSTER [82] using non-crystallographic constraints for complexes 4 and 5 and the 'one TLS group per protein chain' refinement. Structure quality was analysed using *MolProbity* [83]. The anomalous data set from complex 5 was analysed using ANODE [84]. S1 Table summarises the data collection, refinement statistics and the PDB accession codes. All figures showing protein structures (including the porcupine plot in Fig 4A) were generated using *PyMOL* [85]. Motion analysis of protein subdomains was performed using DYNDOM [86].

The structure obtained with the shorter, optimised fragment of ExoY (residues 455 to 863) with the two proteins actin-ATP and profilin (Vn-ExoY-SO$_4$$^{2-}$:actin-ATP-LatB:profilin) is very similar to the structures obtained with the chimera VnE-PRM-Prof (VnE$^{DM}$-PRM-Prof: actin-ADP-LatB or VnE-PRM-Prof-SO$_4$$^{2-}$:actin-ADP, Fig 3A). It shows a similar disorder in the Vn-ExoY switch A, indicating that the switch A dynamics are not induced by the chimera.

The structural rearrangements in the Vn-ExoY switch regions observed between the Vn-ExoY$^{wt}$-SO$_4$$^{2-}$:ATP-actin-LatB:profilin and Vn-ExoY$^{wt}$-3'dATP-2Mg$^{2+}$:ATP-actin-LatB structures are accompanied by minor changes elsewhere in the Vn-ExoY C$_A$ subdomain, such as

helices B (residues 489–504) and I (residues 729–742), which move ~1.8 and ~2.6 Å closer to switch A, respectively (Fig 4A).

## Supporting information

**S1 Table. Table comparing key features of CaM and actin-activated NC toxins in terms of interaction with their specific cofactor, enzymatic specificities and known small-molecule inhibition.** Compared to the Kd of ~0.1–60 nM for CyaA or EF binding to CaM [34,35] or of ~1 μM for Pa-ExoY binding to F-actin [3], Vn-ExoY exhibits only a modest affinity for G-actin (Fig 1D and 1E, Kd~12 μM). The modest affinity of Vn-ExoY is balanced by its ability to interact with and efficiently activate G-actin bound to Profilin (Fig 2A–2C), since the G-actin: Profilin complex is abundant in eukaryotic cells [36,39]. Km, Vmax and Kcat values for different nucleotide substrates and potential NC toxin inhibitors have only been characterized for the calmodulin-activated NC toxins EF and CyaA. The references given are: (Wolff et al. 1980) [91], (Shen et al. 2002) [34], (Gottle et al. 2010) [43], (Soelaiman et al. 2003) [92], (Leppla 1982) [93], (Laine et al. 2010) [94], (Lee et al. 2004) [95], (Belyy et al. 2016) [3], (Becker et al. 2014) [16], (Raoux-Barbot et al. 2018) [5], (Stein, 2022) [35], and (Drum et al., 2000) [96]. (DOCX)

**S2 Table. Crystallographic data-collection and refinement statistics.** The first and last residues indicated in the Vn-ExoY constructs correspond to those of the Uniprot sequence A0A6N3LUE9_9VIBR from the MARTX toxin of *Vibrio nigripulchritudo*, retaining the full numbering. In the text and figures, however, the thousands, i.e. 3000, have been omitted for ease of reading. (DOCX)

**S3 Table. Table showing a comparison of key mechanistic and structural features of the known structures (referenced by their PDB codes) of the CaM-activated EF/CyaA and the actin-activated ExoY NC toxins.** Structures of cofactor-activated NC toxins containing a nucleotide substrate analogue or products of the cyclisation reaction are highlighted in yellow. These structures for Pa-ExoY, EF and CyaA have been compared with the activated Vn-ExoY structures bound to 3'dATP or 3'dCTP. Where one of the mechanistic features of the NBP of Vn-ExoY is not conserved in other NC toxin structures bound to a substrate analogue or to products of the cyclisation reaction, the box in the table corresponding to that feature is coloured red to highlight the differences. No published structures of cofactor-bound NC toxins with a nucleotide substrate analogue of ATP/GTP bound in their NBP exhibit coordination and conformations mechanistically similar or equivalent to the crystal structures of actin-activated Vn-ExoY utilizing 3'dATP or 3'dCTP as bound nucleotide substrate analogues with two metal ions (presented here). (TIF)

**S1 Movie. Rotation of the Vn-ExoY $C_B$/LID subdomain (red) towards its $C_A$ subdomain (black) upon 3'dATP binding between the nucleotide-free Vn-ExoY$^{wt}$-SO42-actin-ATP-LatB:profilin and 3'dATP-bound Vn-ExoY-3'dATP-2*Mg$^{2+}$:actin-ATP-LatB structures.** Side view of both structures superimposed on their black $C_A$ subdomain. The axis of rotation (horizontal in S1 Movie) is indicated by cyan spheres in a line. The $SO_4^{2-}$ ion and 3'dATP-2*Mg$^{2+}$ are *represented* by *ball*-and-*stick models* and their transparent molecular *surfaces*. Secondary structures are shown in yellow, except for those of switch A, B, and C, which are shown in blue, cyan, and purple, respectively. Actin:profilin and actin structures are omitted. Domain motion analysis performed with DYNDOM [86] identifies a 25˚ rotation of the $C_B$/LID subdomain of Vn-ExoY as a rigid body (in red, residues K553-M662$^{VnE-CB/LID}$)

around the hinge regions 532-533$^{VnE}$ and 662-663$^{VnE}$ with respect to the $C_A$ subdomain (in black, residues 468-532$^{VnE-CA}$ and 663-849$^{VnE-CA}$ excluding the switch A, B, C, whose conformations vary between the two structures) (see also Fig 4A and 4B).
(MP4)

**S2 Movie. This video is similar to S1 Movie but provides a frontal view (rotated by 90˚ clockwise) compared to S1 Movie.** The axis of rotation (denoted by a linear arrangement of cyan spheres as in S1 Movie) is here running perpendicular to the figure plane.
(MP4)

**S1 Fig. Trx-Vn-ExoY$^{DM}$ binds free or profilin-bound G-actin with a very similar affinity.** Titration of free or profilin-bound Alexa-488-labelled ATP-G-actin with unlabelled Trx-Vn-ExoY$^{DM}$ and determination of Kd values from changes in microscale thermophoresis (MST) intensities. 0.075 μM Alexa-488-labelled, latrunculin-A-bound G-actin alone/free (cyan) or bound to 4 (red) and 7.9 (green) μM profilin was titrated with Trx-Vn-ExoY$^{DM}$ concentrations as indicated. This corresponds to a 2-fold dilution series from 108 to 0.033 μM with raw data in an insert. Error bars are s.d. (n≥3). The experiment using either the changes in MST (here) or fluorescence (Fig 2C) intensity gave very similar Kd. The Kds averaged over both experiments (MST and fluorescence) were for Trx-Vn-ExoY$^{DM}$ binding to: free G-actin (cyan): Kd = 0.44 ± 0.09 μM), G-actin bound to 4 μM profilin (red): Kd = 0.51 ± 0.12 μM, and G-actin bound to 7.9 μM profilin (green): Kd = 0.43 ± 0.09 μM.
(TIF)

**S2 Fig. Vn-ExoY-induced inhibition of barbed-end growth in the presence of profilin-actin is independent of its adenylate cyclase (AC) activity. A)** Barbed end growth from spectrin-actin seeds (0.22 nM) was measured using 1 μM G-actin (5% pyrenyl-labelled), 8 μM profilin, and the indicated concentrations (μM) of the inactive Trx-Vn-ExoY$^{DM}$ construct. Trx-Vn-ExoY$^{DM}$ inhibits barbed-end elongation from profilin-actin like Vn-ExoY$^{wt}$, demonstrating that Vn-ExoY-mediated effects on actin polymerisation are independent of its AC activity. The buffer was 50 mM KCl, 2 mM MgCl$_2$, 1 mM ATP, 15 mM Tris-HCl pH 7.8, 0.5 mM CaCl$_2$, and 1 mM TCEP. **B)** Barbed end growth from spectrin-actin seeds (0.13 nM) was measured using 1 μM G-actin (5% pyrenyl-labelled), 8 μM profilin, and the indicated concentrations (μM) of active Vn-ExoY$^{wt}$ with either 0.8 (dashed lines) or 2.5 (solid lines) mM ATP. No differences in actin polymerisation kinetics were seen at different ATP concentrations. This indicates that the inhibitory effects of Vn-ExoY on actin polymerisation kinetics are independent of its AC activity throughout the duration of the experiments. Single experiments of actin polymerisation kinetics are shown. They are representative of a minimum of 3 independent experiments.
(TIF)

**S3 Fig. Topology diagrams of (A) Vn-ExoY in PDB: 8BR1, 8BO1 or 8BR0, and (B) Vv-ExoY in PDB 7P1H [28].** (A) The structural topologies correspond in (A) to Vn-ExoY bound to actin-ATP-LatB and 3'dATP-2Mg$^{2+}$ (PDB 8BR1, 2.0 Å resolution), actin-ATP-LatB and 3'dATP-2(Mn/Mg)$^{2+}$ (PDB 8BO1, 2.5 Å resolution) or actin-ADP-LatB and 3'dCTP-2Mn$^{2+}$ (PDB 8BR0, 2.2 Å resolution), and in (B) to Vv-ExoY bound to actin-ATP-LatB and profilin (PDB 7P1H). In this 3.9-Å resolution cryo-EM structure, the 3'dATP ligand used for the preparation of the complex could not be modelled in Vv-ExoY. The amino-acid sequences of the Vn- and Vv-ExoY homologues are 89% similar. However, their structures show a difference in the structural topology of the main chain trace at two segments of their sequence. (C) The superimposed structures of Vn-ExoY and Vv-ExoY, shown in a rainbow of colours (from blue to red from N- to C-terminus, respectively), with their N-terminus enlarged in the right inset.

The backbone trace of Vn-ExoY N-terminal sequence $Y_{468}QSRDLVLEP_{477}$ overlaps with a small region located further in Vv-ExoY backbone. This overlapping region is located between switch B and C (namely between helix H11' and β-stand B9) and corresponds to the Vv-ExoY sequence $L_{332}GEGKGSIQT_{341}$. As a result, the backbone trace of their N-termini (Vn-ExoY sequence $K_{466}TYQSRDLVLEPIQHPKSIEL_{486}$, Vv-ExoY sequence $S_{19}RDLVLEPIVQPE\text{-}TIEL_{34}$) is very different, as is the backbone trace of their region between helix H11' and β-stand B9 (Vn-ExoY sequence $D_{781}DGLGEGKGSIQT_{793}$, Vv-ExoY sequence $E_{329}DGLGEGKGSIQT_{341}$). The ab initio protein structure prediction of Vv-ExoY using the AlphaFold protein-prediction tool [68] suggests that Vv-ExoY adopts a backbone trace topology similar to that observed for Vn-ExoY in the crystal structures. Vv-ExoY bound to actin-ATP:profilin [28] and Vn-ExoY bound to actin-ATP/-ADP and 3'dATP/3'dCTP have otherwise similar overall conformations, including at their switch A, with a r.m.s.d. of 1.4 Å for 364 overlaid Cα atoms. The topology diagrams are adapted from those automatically generated by the Pro-origami [97] and PDBsum [98] programs.
(TIF)

**S4 Fig. Overlays of the Vn-ExoY:3'dATP-2Mg$^{2+}$:actin-ATP-LatB structure (2.1 Å resolution crystal structure, PDB: 8BR1) with (A) the Vv-ExoY:actin-ATP:profilin structure (3.9 Å resolution cryoEM structure, PDB: 7P1H) [28] and (B) Pa-ExoY structure from the Pa-ExoY:3'dGTP-1Mg$^{2+}$:F-actin-ADP-Pi complex (3.2 Å resolution cryoEM structure, PDB: 7P1G) [28].** The structures are superimposed on their C$_A$ subdomain (residues 468-532$^{\text{Vn-ExoY-CA}}$ and 663-849$^{\text{Vn-ExoY-CA}}$). Panels **(A)** and **(B-1)** show the same side views of the complexes, while **(B-2)** is a top view of **(B-1)**.
(TIF)

**S5 Fig. Switch A region is important for G-/F-actin and actin/calmodulin specificity. A)** Docking of the Vn-ExoY-3'dATP-2Mg$^{2+}$:actin-ATP structure in complex with profilin and PRM to F-actin (PDB code: 6FHL) shows that the interaction of the Vn-ExoY switch A between actin subdomains 2 and 4 prevents bound actin (see Fig 4C) from assembling at the barbed (+) end of F-actin. It induces important steric clashes of switch A (red explosion symbols) with the penultimate actin subunit (in pink) at the barbed-end. Switch A binding to actin is also the region most incompatible with longitudinal contacts between adjacent actin subunits of the same F-actin strand (white and pink actin). This also most inhibits Vn-ExoY binding along F-actin [5,28]. **B)** Cartoon representation of the 3'dATP-EF-CaM complex. The NC catalytic domains of EF, CyaA and Vn-ExoY use the same regions, i.e. switch A and C, and a similar orientation to interact with their cofactor. The C-terminal Ca$^{2+}$-binding globular domain of the 8.4-kDa protein CaM is responsible for most of the interactions with the EF or CyaA switch A and C [24,25,27]. However, it is much smaller than 42-kDa actin. It therefore only overlaps with actin subdomains 1 and 3. **C)** Structural comparison of G-actin- and CaM-activated Vn-ExoY and EF conformations, respectively. The common catalytic core, C$_A$ (yellow) and C$_B$ (orange) domains are shown in a cartoon tube representation, and the regions determinant for cofactor specificity, switch A (blue and cyan) and C (pink and purple) regions, are shown in a cartoon representation. The positioning of switch A in EF or CyaA bound to CaM and in Vn-ExoY or Pa-ExoY bound to G or F-actin, respectively, is the most divergent region at the cofactor-toxin binding interface.
(TIF)

**S6 Fig. The structures of EF bound to CaM and ATP analogue (A, B) or Pa-ExoY bound to F-actin and GTP analogue (C) show strong active-site structural heterogeneity.** They show either 1 or 2 metal ions associated with the substrate analogue, variable conformation or

positioning of the different moieties of the purine nucleotide substrate analogues, and different positions of the LID/$C_B$ subdomain relative to the $C_A$ subdomain. The protein structures are shown in cartoon and coloured in cyan (PDB: 1XFV [27]), green (PDB: 1K90 [24]), light magenta (PDB: 1S26 [29]) and yellow (PDB: 7P1G [28]). The catalytic domains of the EF and Pa-ExoY NC toxins are superimposed on their $C_A$ subdomain. Metal ions are shown as spheres and the ligands and side chains of important catalytic residues (K346[EF-CA]/K81[Pa-ExoY-CA], H353[EF-LID]/K88[Pa-ExoY-LID] and H577[EF-Switch-B]/H292[Pa-ExoY-Switch-B]) are shown as sticks. **(A)** Overlay of the CaM-activated structures of EF bound to 3'dATP containing 1 or 2 metal cofactors. **(B)** Overlay of the CaM-activated structures of EF bound to ATP analogues containing 1 metal cofactor. **(C)** Overlay of EF CaM-activated structure bound to 3'dATP with 1 Yb$^{3+}$ metal ion from (A and B) and Pa-ExoY F-actin-activated structure bound to 3'dGTP with 1 Mg$^{2+}$ metal ion. S7 Fig complements S6 Fig, offering together unambiguous evidence of the variations in the number of ions, the position and coordination of metal ion(s), and the position, conformation, and coordination of nucleotide substrate analogues. This includes variations in the coordination between the enzyme and nucleotide, as well as between the nucleotide and the metal ion(s). For instance, panel S6-B illustrates that the adenine ring of the AMPCPP bound to CaM-bound EF (PDB 1S26) is rotated approximately 180° compared to 3'dATP bound to CaM-bound EF (PDB 1XFV, 1K90) concerning the nucleotide conformation. Moreover, panels S6-A and S6-C demonstrate that 3'dATP (S6A Fig) or 3'dGTP (S6B Fig) can be bound with either 1 or 2 metal ions in previous structures. Additionally, the conserved K353EF-LID (S6A Fig) or K88PaE-LID (S6C Fig), equivalent to K535VnE-LID in Vn-ExoY (Figs 6A and 7A), can either be away from the γ-phosphate oxygens or coordinate with them.

(TIF)

**S7 Fig. Interactions with non-cyclisable ATP or GTP analogues and metal ions, and position of the LID/$C_B$ relative to the $C_A$ subdomain in CaM-bound EF/CyaA and F-actin-bound Pa-ExoY structures.** The active site of **(A)** CaM-activated EF with 3'dATP and two Mg$^{2+}$ metal ions (3.35 Å resolution crystal structure, PDB: 1XFV) [25], **(B)** CaM-activated CyaA structure with adefovir diphosphate (9-(2-(phosphonomethoxy)ethyl)adenine diphosphate, or PMEAPP) and two Mg$^{2+}$ metal ions (2.20 Å resolution crystal structure, PDB: 1ZOT) [27], **(C)** F-actin-activated Pa-ExoY with 3'dGTP and a single Mg$^{2+}$ metal ion (cryoEM structure at an average 3.20 Å resolution, PDB: 7P1G) [28], **(D)** CaM-activated EF structure with 3'dATP and a single Yb$^{3+}$ metal ion (2.75 Å resolution crystal structure, PDB: 1K90) [24], and **(E)** G-actin-activated Vn-ExoY with 3'dATP and two Mg$^{2+}$ metal ions (2.04 Å resolution crystal structure presented in this article, PDB: 8BR1). Thresholds for interaction detection are those of the PLIP (protein-ligand interaction profiler) [88] and Arpeggio [89] web servers, and those from the PoseView [99] tool available on the ProteinsPlus web server [90,100]. S8 Fig shows the position of the LID/$C_B$ relative to the $C_A$ subdomain in PDBs 1XFV (A), 1ZOT (B) and 1K90 (D). The use of an unconventional metal ion such as Yb$^{3+}$ for CaM-bound EF crystallisation is expected to alter the coordination of EF NBP with ATP only slightly compared to the putative physiological metal-ligand Mg$^{2+}$, as the metal Yb$^{3+}$ only slightly reduces the AC catalytic activity of EF [24].

(TIF)

**S8 Fig. The LID/$C_B$ subdomain of CaM-activated EF and CyaA structures, in complex with various non-cyclisable ATP analogues, adopts different positions relative to the $C_A$ subdomain [24, 25], resulting in different positions and interactions with the ATP analogues and metal-ion(s) as shown in S6 and S7 Figs.** The EF structure with the LID in a closed conformation (1K90) and the EF and CyaA structures with the LID in an open conformation

(1XFV and 1ZOT, respectively) are superimposed on their $C_A$ subdomains (in yellow). Protein domain motion analysis between the two structures of EF bound to CaM and 3'dATP (PDB 1K90 and 1XFV) shows that the LID/$C_B$ subdomain undergoes a 16° rotation as a rigid body around hinge regions 350-351$^{EF}$ and 488-489$^{EF}$ (motion analysis performed with DYNDOM [86]). Switch A, B, and C regions of EF and CyaA are coloured in blue, cyan, and purple, respectively. The LID region is shown in orange (1K90), pink (1XFV), and red (1ZOT). CaM and the additional C-terminal helical domain of EF are shown in white and green, respectively. Switch A exhibits minimal stabilisation (or only pre-stabilisation) by actin in the absence of substrate (Figs 3D–3F and 4D). On the other hand, EF and CyaA present a fully folded switch A in their nucleotide-free, CaM-bound state [24,25]. However, their $C_B$/LID subdomain displays flexibility and can adopt various positions in isolated CyaA [53] or in different crystal structures of CaM-bound EF (as shown here and in Fig 6E). Numerous crystal structures of CaM-activated EF or CyaA, with nucleotide ligands bound into their active site, were crystallised without substrate analogues or reaction products and subsequently exposed to these ligands. The crystal packing in these structures may have hindered the complete rearrangement of the NBP and LID/$C_B$ subdomain closure (S3 Table). Consequently, due to the inherent flexibility of the LID/$C_B$ subdomain, both substrate analogues and metal ions occupy significantly distinct positions in the NBP of CaM-bound EF or CyaA (S6–S8 Figs) [1]. These distinct structures likely correspond to intermediate stages of LID/$C_B$ movement and substrate entry.
(TIF)

**S9 Fig. Binding interface overlap between the binding of Vn-ExoY switch C and the binding of major G-actin-binding (GAB) proteins/domains: (A) Thymosin-β4 (Tβ4) (PDB: 4PL7 [101]), (B) ADF (PDB: 3DAW [102]), (C) Cordon-bleu WH2 domain (PDB: 5YPU [103]), (D) GAB domain of VASP (PDB: 2PBD [79]).** The interaction of the C-terminal amphipathic α-helix of Vn-ExoY switch-C within the hydrophobic cleft between actin subdomains 1 and 3 (shown in the right zoom panel) is a common binding site of actin-binding proteins (ABPs) [104]. Apart from profilin, this interaction of the Vn-ExoY switch-C C-terminus competes with a similar interaction that exists with most G-ABPs as shown in these examples. The structural models are consistent with Fig 2A in the main text, which shows the competition in solution between Vn-ExoY and the Tβ4 or Cordon-bleu WH2 domain for binding to G-actin. Taken together, these data suggest that the G-actin:profilin complex serves as a physiological cofactor for Vn-ExoY and its closely-related ExoYs (Figs 5D and S10) in eukaryotic cells.
(TIF)

**S10 Fig. Phylogenetic tree showing the relationships between ExoY-like proteins/effector domains produced by different β- and γ-proteobacteria.** ExoYs are classified as Vn-ExoY-like or Pa-ExoY-like homologues based on both their sequence similarity and predicted structure. Except for the indicated PDBs, the *ab initio* protein structure of all other ExoYs was predicted using the highly-accurate protein prediction tool AlphaFold [68]. The amino-acid sequence alignment of Pa-ExoY, Vn-ExoY and ExoY-like proteins/modules found in other γ- or β-proteobacteria and potentially activated by actin was performed using Clustal Omega (http://www.ebi.ac.uk, accessed on 12 September 2022) [105]. The multiple sequence alignment from Clustal Omega was used to construct the phylogenetic tree by using TreeDyn (www.phylogeny.fr) [106]. Pairwise sequence similarities (%) with Pa-ExoY and Vn-ExoY are shown in green and blue, respectively (Sequence Identity And Similarity (SIAS) tool, http://imed.med.ucm.es/Tools/sias.html accessed 12 September 2022). The NCBI accession numbers of the protein sequences are given to the right of the phylogenetic tree. The structures of Vn-

ExoY solved in this study with actin or actin:profilin are marked with a red asterisk. The selectivity of the switch A conformation for G- or F-actin in the solved or AlphaFold-predicted structures of the ExoYs is indicated by G or F, respectively. The predicted structures of the ExoYs, which are closely-related to the MARTX Vn-ExoY module, are compatible with the formation of an ExoY:actin:profilin ternary complex and a conformation of switch A on actin that inhibits G-actin:profilin assembly at the barbed-ends of actin filaments. A + sign indicates that the Vn-ExoY or Pa-ExoY residue/structural element is conserved at the same position in the ExoY-like homologue's sequence and predicted structure.
(TIF)

**S11 Fig.** Protein-ligand interactions of the ATP analogue and metal ions **in the active site of Vn-ExoY class II AC (A, PDB: 1BR1) and mammalian transmembrane class III AC (tmAC) (B, PDB: 1CJK) activated by their respective cofactors are shown. (A-1)** A simplified model of ATP binding in activated Vn-ExoY is presented, based on its interactions with 3'dATP in the Vn-ExoY-3'dATP-2*$Mg^{2+}$:ATP-actin-LatB structure. **(A-2)** Three-dimensional protein-ligand interactions in the corresponding crystal structure. **(B-1)** A simplified model of ATP binding in activated class III tmAC is presented, based on the structure of the type V AC C1a/type II AC C2 heterodimer bound to the ATP analogue RP-ATPαS (PDB 1CJK). **(B-2)** Three-dimensional protein-ligand interactions in the corresponding crystal structure. For clarity and simplicity, only a few residues crucial for the purinylyl cyclase activities of Vn-ExoY and tmAC are shown here, while the full protein-ligand interactions are illustrated in Fig 6A and 6F. The thresholds used for interaction detection follow the criteria of the PLIP [11] and Arpeggio [12] web servers, along with the PoseView tool in the ProteinsPlus web server [14,15]. Regarding the catalysis mechanism of tmAC, a computational study [42] has proposed a substrate-assisted general base catalysis, wherein several residues of the enzyme with metal-ion cofactors play essential roles in reducing the two highest energy barriers of the reaction. Specifically, K1065[tmAC-IIC2], $Mg^{2+}_A$, $Mg^{2+}_B$ and the two conserved D96D440[tmAC-VC1] anchoring metal ions are critical in facilitating the proton transfer from the ribosyl 3'O to a γ-phosphate oxygen [42]. Subsequently, R1029[tmAC-IIC2] is identified as critical for the concerted phosphoryl transfer step, with K1065[tmAC-IIC2] also involved. Further investigation is required to determine the significance of H753[VnE-switch-B] (or its equivalent histidine in other NC toxins) in initiating 3'OH deprotonation via $Mg^{2+}_A$ and whether K528[VnE-CA] (or equivalent) is essential for the phosphoryl transfer step, similar to R1029[tmAC-IIC2] in tmAC.
(TIF)

**S12 Fig. SDS-PAGE and Coomassie Blue staining of proteins used in functional assays.** Molecular mass markers (in kDa) are shown on the left A–C (M, lane 1). (A), lanes 2–10: purified actin, chimera Vn-ExoY (wt/DM)-PRM-Prof, Vn-ExoY[wt] (residues 455–863), Trx-Vn-ExoY[DM], Vn-ExoY mutants (residues 455–863): K528Q[VnE-CA], K533Q[VnE-LID], K535Q[VnE-LID], H753Q[VnE-switch-B], respectively. (B), lanes 2–6: Trx-Vn-ExoY[wt,] Vn-ExoY[wt] (residues 455–863), Vn-ExoY mutants (residues 455–863): R824A[VnE-switch-C], 5 Mut (Y850A-R854A-V857A-K860A-L861A)[VnE-switch-C] and actin, respectively. (C), lanes 2–10: Pa-ExoY wt/Mutants (residues 20 to 378): Pa-ExoY[wt], 3 Mut (F367A-L371A-F374A)[PaE-switch-C], Pa-ExoY K81Q[PaE-CA], K86Q[PaE-Lid], K88Q[PaE-LID], H291Q[PaE-switch-B], H352A[PaE-switch-C], K347A[PaE-switch-C], R364A[PaE-switch-C], respectively. All samples were centrifuged at 16,000 g for 10 min before loading 3.5 μM, separating by 10% SDS-PAGE and staining with Coomassie blue.
(TIF)

**S13 Fig. Elution profiles of actin and the main constructs of Vn-ExoY and Pa-ExoY used in this study, by size exclusion chromatography (SEC) on a Superdex 200 HR 10/300 column.**

Elution profiles of (**A**-**B**) actin (42 kDa) and purified Vn-ExoY$^{wt/mutant}$ proteins (residues 455–863, 50 kDa), and (**C**-**D**) actin and purified Pa-ExoY $^{wt/mutant}$ proteins (residues 20 to 378, 44 kDa). The proteins are: (**A**) Latrunculin B-bound actin-ATP (elution volume of 14.79 mL), Vn-ExoY mutant K533Q$^{VnE-LID}$ (13.76 mL), the chimera Vn-ExoY$^{DM}$-PRM-Prof (72 kDa, 13.17 mL), Vn-ExoY mutants: H753Q$^{VnE-switch-B}$ (13.80 mL), 5 Mut (Y850A-R854A-V857A--K860A-L861A) (13.70 mL) and Vn-ExoY$^{wt}$ (13.81 mL), (**B**) actin-ATP-LatB (14.79 mL), Vn-ExoY mutants K528Q$^{VnE-CA}$ (13.79 mL), K535Q$^{VnE-LID}$ (13.77 mL), Trx-Vn-ExoY$^{DM}$ (69 kDa, 13.20 mL), Vn-ExoY mutant R824$^{VnE-switch-}$C (13.77 mL) and Vn-ExoY$^{WT}$ (13.81 mL), (**C**) Pa-ExoY mutants: K347A$^{PaE-switch-C}$, R364A$^{PaE-switch-C}$, Pa-ExoY$^{wt}$, Pa-ExoY mutant K86Q$^{PaE-LID}$ and actin-ATP-LatB (14.79 mL), (**D**) Pa-ExoY mutants: 3 Mut (F367A-L371A-F374A), H291Q$^{PaE-switch-B}$, H352A$^{PaE-switch-C}$, Pa-ExoY$^{wt}$, Pa-ExoY mutants: Pa-ExoY K88Q$^{PaE-LID}$, K81Q$^{PaE-LID}$ and actin-ATP-LatB (14.79 mL). All the Pa-ExoY $^{wt/mutant}$ proteins were eluted with a similar elution volume of 14.74 mL. The purified proteins were ultracentrifuged at 100,000 g for 20 min at 4˚C and loaded as a volume of 200 μl at a concentration between 2 and 5 μM to distinguish between very similar elution profiles. Protein elution at 4˚C was monitored by absorbance at 280 nm on a Superdex 200 HR 10/300 SEC column (~24 mL) equilibrated in (25 mM Tris-HCl pH 7.5, 150 mM KCl, 2 mM MgCl$_2$ and 1 mM DTT). V$_0$ is the void/dead volume of the column (7.9 mL). The elution volumes of the proteins used for column calibration are shown in brackets and indicated by black arrows at the top of each panel. These include catalase (Cat, 232 kDa, 12. 5 mL), bovine serum albumin (BSA, 67 kDa, 13.6 mL), ovalbumin (Ova, 44 kDa, 14.6 mL) and ribonuclease (RNase A, 13.7 kDa, 16.9 mL).
(TIF)

**S1 Data. Excel spreadsheet containing, on separate sheets, the underlying numerical data that were used to generate plots or histograms for Figs 1A–1D, 1E—left panel, 1E—right panel, 2A-2F, 3B, 3C, 5A, 5C, 6C, 6D, 7B, 7C, S1, S2A, S2B, S13A–S13D.**
(XLSX)

## Acknowledgments

We thank SOLEIL for providing synchrotron radiation facilities, PROXIMA-1 and PROXIMA-2 beamline staff, and Pierre Legrand for strong technical support during data collection. We acknowledge the core facilities of Imagerie-Gif (http://www.i2bc.paris-saclay.fr). The core facilities of Imagerie-Gif were supported by "France-BioImaging" (ANR-10-INBS-04-01). This work has benefited from I2BC's crystallization and protein-protein interaction platforms, supported by French Infrastructure for Integrated Structural Biology (FRISBI) ANR-10-INBS-05.

## Author Contributions

**Conceptualization:** Louis Renault.

**Formal analysis:** Magda Teixeira Nunes, Pascal Retailleau, Dorothée Raoux-Barbot, Martine Comisso, Anani Amegan Missinou, Christophe Velours, Undine Mechold, Louis Renault.

**Funding acquisition:** Daniel Ladant, Undine Mechold, Louis Renault.

**Investigation:** Magda Teixeira Nunes, Pascal Retailleau, Dorothée Raoux-Barbot, Martine Comisso, Anani Amegan Missinou, Christophe Velours, Stéphane Plancqueel, Undine Mechold, Louis Renault.

**Project administration:** Louis Renault.

**Resources:** Daniel Ladant, Undine Mechold, Louis Renault.

**Supervision:** Undine Mechold, Louis Renault.

**Validation:** Magda Teixeira Nunes, Pascal Retailleau, Undine Mechold, Louis Renault.

**Visualization:** Magda Teixeira Nunes, Dorothée Raoux-Barbot, Martine Comisso, Christophe Velours, Louis Renault.

**Writing – original draft:** Magda Teixeira Nunes, Louis Renault.

**Writing – review & editing:** Magda Teixeira Nunes, Pascal Retailleau, Martine Comisso, Anani Amegan Missinou, Christophe Velours, Daniel Ladant, Undine Mechold, Louis Renault.

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
