## [Decision Letter · Decision Letter 0]

2 Jun 2023

Dear Dr RENAULT,

Thank you very much for submitting your manuscript "Functional and structural insights into the multi-step activation and catalytic mechanism of bacterial ExoY nucleotidyl cyclase toxins bound to actin-profilin" for consideration at PLOS Pathogens. As with all papers reviewed by the journal, your manuscript was reviewed by members of the editorial board and by several independent reviewers. 

The article presents a number of results of interest and unquestionable quality, as clearly stated by all reviewers, not requiring any additional experiments. However, as also mentioned by Reviewers 1 and 2, the article discussion is extremely long and contains sections with somewhat redundant descriptions of results and whose removal from the discussion or simplification could help a more fluent and understandable reading for readers not so familiar with the topic. Thus, the authors should revise the article in order to implement this simplification. For this, they can follow the recommendations suggested by the reviewers or do it in some alternative way according to their preference. They may, eventually, consider passing some of the details they consider more specific to the supplementary information, incorporating the more detailed discussion in the captions of the respective supplementary figures or creating an “extended discussion” section. Additionally, authors should respond/incorporate into the article other major issues raised by reviewers, such as clearly mentioning the number of technical or biological replicates used, when appropriate, and the simplification of some figure panels, as much as possible (please use the supplementary information as support whenever necessary).

Thus, based on the reviews, we are likely to accept this manuscript for publication, providing that you modify the manuscript according to the review recommendations.

Sincerely,

Nuno M. S. dos Santos

Academic Editor

PLOS Pathogens

David Skurnik

Section Editor

PLOS Pathogens

Kasturi Haldar

Editor-in-Chief

PLOS Pathogens

orcid.org/0000-0001-5065-158X

Michael Malim

Editor-in-Chief

PLOS Pathogens

orcid.org/0000-0002-7699-2064

Reviewer Comments (if any, and for reference):

Reviewer's Responses to Questions

**Part I - Summary**

Reviewer #1: This is a superb paper of a world-leading research consortium describing the structural basis for the catalytic mechanism of ExoY-like bacterial toxins. These toxins are unique in the sende that they use ATP, GTP, CTP and UTP as substrates. The authors use a broad array of biophysical and biochemical methods to delineate the molecular basis of catalysis of these toxins. The dimensions of this paper are titanic, i.e., the authors could have easily written two papers, but they refrained from doing so. The most spectalular finding is the presentation of an ExoY-like toxin in complex with the pyrimidine nucleotide 3'dCTP.

Reviewer #2: The manuscript by Teixeira-Nunes et al describes how a nucleotidyl cyclase ExoY, Vn-ExoY, interacts with monomeric actin. The authors show that Vn-ExoY can bind G-actin-profilin complex, describe a series of X-ray structures of the complex in the absence of ATP, or bound to 3’-deoxy-ATP or 3’deoxy-CTP (ATP analogues), revealing that substrate binding leads to a closure of the nucleotide binding pocket. The authors suggest that the ExoY-like toxins utilise a two metal ion catalytic reaction and propose a mechanism of Vn-ExoY activation mediated by actin-profilin binding.

This study includes a lot of interesting experimental results and the quality of the experiments appears to be very good. The topic is very important, and the authors did a great job addressing the mechanism of NC toxin activation. In principle the results and interpretations are clear and the story presented by the authors is well supported by the data. However, the quality of the manuscript can be substantially improved by updating the figures and the discussion. There is a lot of redundancy in the discussion and the text can be greatly optimised. The quality of the figures has to be improved - many of the figure panels are very dense, making them rather hard to follow.

I would consider this a "major minor revision". All of my comments can be considered minor, but I grade some of them as major based on the effect that these points have on the presentation of the data. I have no doubt that the authors will be able to improve the figures and the text accordingly.

Reviewer #3: Please see attached review.

**Part II – Major Issues: Key Experiments Required for Acceptance**

Reviewer #1: The paper contains so many interesting data that it is almost unfair to ask for more. However, the length of the paper reduces its accessibility for non-experts.

I have the following suggestions for revision:

1. Please, add subsections to the Discussion.

2. Please, emphasize more clearly the novelty of your ExoY-3'dCTP structure.

3. What about the Km, Vmax and Kcat values for all NTPs?

4. What about potential compounds inhibiting ExoY?

5. It would be most helpful to have a short summary table for the discussion comparing the most important features of ExoY, EF and CyaA.

Reviewer #2: Discussion: a lot of the “Discussion” space is used to repeat the findings in the “Results” part, with multiple references to figures. Without a doubt the authors can improve the discussion by focusing on the important points, not repeating the results again and improving the accompanying illustration (Fig 8, see below).

Fig. 4A-B. There is a lot going on here in each of the panels. I feel that the figure could be simplified by carefully selecting the most important things that the reader should look at and colouring those - reducing the complexity as much as possible, while keeping the message clear. Any solution could work as long as it reduces the impression of cacophony currently present in these two figure panels.

Fig. 4D. Similarly, it may be a good idea to reduce the complexity of the figure panel D. The fewer unnecessary elements are shown, the easier it is to follow what the authors are presenting.

Fig. 6A - is it necessary to have such a complicated presentation in a main figure? Can this panel be simplified? I think it should be possible to have a panel that conveys the main ideas, but with fewer details shown all at once.

Fig. 6A and 6F. Although this is an interesting schematic representation of the nucleotide binding site comparison between the Eco and the Gs-activated mAC, it would be more informative to show an example of the main residues in an active site of an actual mAC structure, compared with the active site in ExoY. It would be informative to see how active sites are organised in class II and class III cyclases, and how the new structures reported here advance our understanding of the similarities / differences.

Fig. 8. The scheme is very complicated, and since there are no indices to follow, it is actually hard to decipher what is shown. The authors can try and optimise this figure by keeping a uniform style of presentation for different items shown in this figure (currently a rather eclectic mixture of different styles), and by using indices (such as “1, 2, 3, ..” or “A, B, C, ..”) to help convey the flow of what is shown and to guide the reader through the mechanism.

Reviewer #3: See attached review.

**Part III – Minor Issues: Editorial and Data Presentation Modifications**

Reviewer #1: (No Response)

Reviewer #2: Supporting Information: please add protein biochemistry data SDS PAGE, gel filtration profiles, etc. Although it is clear that the X-ray structures have been determined at high resolution, it makes sense to see the purity of the samples used in functional assays.

Fig. 1-3. How many times were the experiments repeated? Is n in Fig. 2B a number of replicates or repeats? Number of repeats should be specified for each figure panel where experimental data is presented.

Fig. 3C. Growth rates are plotted against the toxin concentration, but “Kd” values are shown. What does this mean? What is “Kd” in this case? Although it is clear that Kd should be the dissociation constant in a binding assay, the meaning of Kd here is less clear.

Fig. 4D. In the transition from D-1 to D-2 - is that ATP molecule above the arrow? Or an ATP analogue?

Fig. S2. How many times were these experiments performed?

Fig. S3C. There might be a problem with the labels.

Fig. S6. The overlayed structures are very crowded - as a result the active site heterogeneity does not immediately come across. In fact, the active site itself appears to not be all that heterogeneous.

Fig. S8-S9 - the resolution of the figure can be improved.

Reviewer #3: See attached review.

PLOS authors have the option to publish the peer review history of their article (what does this mean?). If published, this will include your full peer review and any attached files.

Reviewer #1: No

Reviewer #2: No

Reviewer #3: No

Figure Files:

Data Requirements:

Reproducibility:

References:

---

## [Decision Letter · Decision Letter 1]

1 Sep 2023

Dear Dr RENAULT,

We are pleased to inform you that your manuscript 'Functional and structural insights into the multi-step activation and catalytic mechanism of bacterial ExoY nucleotidyl cyclase toxins bound to actin-profilin' has been provisionally accepted for publication in PLOS Pathogens.

Best regards,

Nuno M. S. dos Santos

Academic Editor

PLOS Pathogens

David Skurnik

Section Editor

PLOS Pathogens

Kasturi Haldar

Editor-in-Chief

PLOS Pathogens

orcid.org/0000-0001-5065-158X

Michael Malim

Editor-in-Chief

PLOS Pathogens

orcid.org/0000-0002-7699-2064

Reviewer Comments (if any, and for reference):

Reviewer's Responses to Questions

**Part I - Summary**

Reviewer #1: The paper makes an important contribution to our understanding of bacterial nucleotidyl cyclase toxins.

Reviewer #2: The revised manuscript is substantially improved. The authors have addressed all comments in a satisfactory manner.

Reviewer #3: The authors have satisfactorily addressed the issues that I raised in my initial review. The additional materials provided as supplementary information are useful additions, and the discussion has been greatly improved. I recommend publication of the manuscript as revised.

**Part II – Major Issues: Key Experiments Required for Acceptance**

Reviewer #1: The authors have addressed all of my suggestions.

Reviewer #2: There are no major issues.

Reviewer #3: (No Response)

**Part III – Minor Issues: Editorial and Data Presentation Modifications**

Reviewer #1: all aspects have been considered.

Reviewer #2: My only suggestion here is to carefully go through each figure - main and supplementary - and check for pixelation. The figure versions I saw in this submission, especially the structure figure panels, are in places strongly pixelated. In some panels the effect is less pronounced (e.g., Fig. S5 - strongly pixelated; S11 - also pixelated, but the effect is less apparent). If the authors used PyMol for generating these figures, they should consider in each case to reproduce the figure panel with an additional command "ray 2000". This will generate high resolution versions of these figures.

Reviewer #3: (No Response)

PLOS authors have the option to publish the peer review history of their article (what does this mean?). If published, this will include your full peer review and any attached files.

Reviewer #1: No

Reviewer #2: No

Reviewer #3: No

---

## [Editor Report · Acceptance letter]

20 Sep 2023

Dear Dr RENAULT,

We are delighted to inform you that your manuscript, "Functional and structural insights into the multi-step activation and catalytic mechanism of bacterial ExoY nucleotidyl cyclase toxins bound to actin-profilin," has been formally accepted for publication in PLOS Pathogens.

Best regards,

Kasturi Haldar

Editor-in-Chief

PLOS Pathogens

orcid.org/0000-0001-5065-158X

Michael Malim

Editor-in-Chief

PLOS Pathogens

orcid.org/0000-0002-7699-2064